

# Inter-annual variability of the global terrestrial water cycle

Dongqin Yin[1,2], Michael L. Roderick[1,2,3]

[1]Research School of Earth Sciences, Australian National University, Canberra, ACT, 2601, Australia

[2]Australian Research Council Centre of Excellence for Climate System Science, Canberra, ACT, 2601, Australia

[3]Australian Research Council Centre of Excellence for Climate Extremes, Canberra, ACT, 2601, Australia

*Correspondence to*: (dongqin.yin@anu.edu.au)

**Abstract:**

Variability of the terrestrial water cycle, i.e., precipitation ($P$), evapotranspiration ($E$), runoff ($Q$) and water storage
change ($\Delta S$) is the key to understanding hydro-climate extremes. However, a comprehensive global assessment
for the partitioning of variability in $P$ between $E$, $Q$ and $\Delta S$ is still not available. In this study, we use the recently
released global monthly hydrologic reanalysis product known as the Climate Data Record (CDR) to conduct an
initial investigation of the inter-annual variability of the global terrestrial water cycle. We first examine global
patterns in partitioning the long-term mean $\bar{P}$ between the various sinks $\bar{E}$, $\bar{Q}$ and $\overline{\Delta S}$ and confirm the well-known
patterns with $\bar{P}$ partitioned between $\bar{E}$ and $\bar{Q}$ according to the aridity index. In a new analysis based on the concept
of variability source and sinks (Eq. 2) we then examine how variability in the precipitation $\sigma_P^2$ (the source) is
partitioned between the three variability sinks $\sigma_E^2$, $\sigma_Q^2$ and $\sigma_{\Delta S}^2$ along with the three relevant covariance terms, and
how that partitioning varies with the aridity index. We find that the partitioning of inter-annual variability does
not simply follow the mean state partitioning, with $\sigma_P^2$ mostly partitioned between $\sigma_Q^2$, $\sigma_{\Delta S}^2$ and the associated
covariances. We also find that the magnitude of the covariance components can be large and often negative,
indicating the variability in the sinks (e.g., $\sigma_Q^2$, $\sigma_{\Delta S}^2$) can, and do, exceed variability in the source ($\sigma_P^2$). Further
investigations under extreme conditions reveal that in extremely dry environments the variance partitioning is
closely related to the water storage capacity. With limited storage capacity the partitioning of $\sigma_P^2$ is mostly to $\sigma_E^2$,
but as the storage capacity increases the partitioning of $\sigma_P^2$ is increasingly shared between $\sigma_E^2$, $\sigma_{\Delta S}^2$ and the
covariance between those variables. In other environments (i.e., extremely wet and semi-arid/semi-humid) the
variance partitioning proved to extremely complex and a synthesis was not developed. We anticipate that a major
scientific effort will be needed to develop a synthesis of hydrologic variability.



## 1. Introduction

In describing the terrestrial branch of the water cycle, the precipitation ($P$) is partitioned into evapotranspiration ($E$), runoff ($Q$) and change in water storage ($\Delta S$). With averages taken over many years, $\overline{\Delta S}$ is usually assumed to be zero and it has long been recognized that the partitioning of the long-term mean annual precipitation ($\bar{P}$) between $\bar{E}$ and $\bar{Q}$ was jointly determined by the availability of both water ($\bar{P}$) and energy (represented by the net radiation expressed as an equivalent depth of water and denoted $\overline{E_o}$) fluxes. Using data from a large number of watersheds, Budyko (1974) developed an empirical relation relating the evapotranspiration ratio ($\bar{E}/\bar{P}$) to the aridity index ($\overline{E_o}/\bar{P}$). The resultant empirical relation and other Budyko-type forms (e.g., Fu, 1981; Choudhury, 1999; Yang et al., 2008, Roderick and Farquhar, 2011; Sposito, 2017) that partition $P$ between $E$ and $Q$ have proven to be extremely useful in both understanding and characterising the long-term mean annual hydrological conditions in a given region.

However, the long-term mean annual hydrologic fluxes rarely occur in any given year. Instead, society must (routinely) deal with variability around the long-term mean. The classic hydro-climate extremes are droughts and floods but the key point here is that hydrologic variability is expressed on a full spectrum of time and space scales. To accommodate that perspective, we need to extend our thinking beyond the long-term mean to ask how the variability of $P$ is partitioned into the variability of $E$, $Q$ and $\Delta S$?

Early research on hydrologic variability focussed on extending the Budyko curve. In particular, Koster and Suarez (1999) used the Budyko curve to analyse inter-annual variability in the water cycle. In their framework, the evapotranspiration standard deviation ratio (defined as the ratio of standard deviation for $E$ to $P$, $\sigma_E/\sigma_P$) was (also) estimated using the aridity index ($\overline{E_o}/\bar{P}$). The classic Koster and Suarez framework has been widely applied and extended in investigations of the variability in both $E$ and $Q$, using catchment observations, reanalysis data and model outputs (e.g., McMahon et al., 2011; Wang and Alimohammadi 2012; Sankarasubramanian and Vogel, 2002; Zeng and Cai, 2015). However, typical applications of the Koster and Suarez framework have previously been at regional scales and there is still no comprehensive global assessment for the partitioning of variability of $P$ into the variability of $E$, $Q$ and $\Delta S$. One reason for the lack of a global comprehensive assessment is the absence of gridded global hydrologic data. Interestingly, the atmospheric science community have long used a



combination of observations and model outputs to construct global atmospheric re-analyses and such products
have become central to atmospheric research. Those atmospheric products also contain estimates of some of the
key water cycle variables (e.g., $P$, $E$), such as in the widely used interim ECMWF Re-Analysis (ERA-Interim;
Dee et al. 2011). However, the central aim of atmospheric re-analysis is to estimate atmospheric variables, which,
understandably, ignores many of the nuances of soil water infiltration, vegetation water uptake, runoff generation
and many other processes of central importance in hydrology.

Hydrologists have only recently accepted the challenge of developing their own re-analysis type products with
perhaps the first serious hydrologic re-analysis being published as recently as a few years ago (Rodell et al., 2015).
More recently, the Princeton University group has extended this early work by making available a gridded global
terrestrial hydrologic re-analysis product known as the Climate Data Record (CDR) (Zhang et al., 2018). Briefly,
the CDR was constructed by synthesizing multiple in-situ observations, satellite remote sensing products, and
land surface model outputs to provide *gridded* estimates of global land precipitation $P$, evapotranspiration $E$,
runoff $Q$ and total water storage change $\Delta S$ ($0.5° \times 0.5°$, monthly, 1984-2010). In developing the CDR, the authors
adopted local water budget closure as the fundamental hydrologic principle. That approach presented one
important difficulty. Global observations of $\Delta S$ start with the GRACE satellite mission from 2002. Hence before
2002 there is no direct observational constraint on $\Delta S$ and the authors made the further assumption that the mean
annual $\Delta S$ over the full 1984-2010 period was zero at every grid-box. That is incorrect in some regions (e.g.
Scanlon et al., 2018) and represents an observational problem that cannot be overcome. However, our interest is
in the year-to-year variability and for that application, the assumption of no change in the mean annual $\Delta S$ over
the full 1984-2010 period is unlikely to lead to major problems since we are not looking for subtle changes over
the full time series. With that caveat in mind, the aim of this study is to use this new 27-year gridded hydrologic
re-analysis product to conduct an initial investigation of the inter-annual variability of the terrestrial branch of the
global water cycle.

The paper is structured as follows. We begin in Section 2 by describing the various climate and hydrologic
databases including a further assessment of the suitability of the CDR database for this initial variability study. In
Section 3, we examine relationships between the mean and variability in the four water cycle variables ($P$, $E$, $Q$
and $\Delta S$). In Section 4, we first relate the variability to classical aridity index and then use those results to evaluate



the theory of Koster and Suarez (1999). Subsequently we examine how the variance of $P$ is partitioned into the
variances (and relevant covariances) of $E$, $Q$ and $\Delta S$ and investigate some factors controlling the variance
partitioning. We finalise the paper with a discussion summarising what we have learnt about water cycle
variability over land by using the CDR database.

**2. Methods and Data**
2.1 Methods
The water balance is defined by,
$$P(t) = E(t) + Q(t) + \Delta S(t) \qquad (1)$$
with $P$ the precipitation, $E$ the evapotranspiration, $Q$ the runoff and $\Delta S$ the total water storage change in time
step $t$. By the usual variance law, we have,
$$\sigma_P^2 = \sigma_E^2 + \sigma_Q^2 + \sigma_{\Delta S}^2 + 2cov(E,Q) + 2cov(E,\Delta S) + 2cov(Q,\Delta S) \qquad (2)$$
that includes all relevant variances (denoted $\sigma^2$) and covariances (denoted $cov$). Eq. (1) is the familiar
hydrologic mass balance equation. In that context, Eq. (2) can be thought of as the hydrologic variance balance
equation.

2.2 Hydrologic and Climatic Data

We use the recently released global land hydrologic re-analysis known here as the Climate Data Record (CDR)
(Zhang et al., 2018). This product includes global precipitation $P$, evapotranspiration $E$, runoff $Q$ and water storage
change $\Delta S$ (0.5° × 0.5°, monthly, 1984-2010). The CDR does not report additional radiative variables and we use
the NASA/GEWEX Surface Radiation Budget (SRB) Release-3.0 (monthly, 1984-2007, 1° × 1°) database
(Stackhouse et al., 2011) to calculate $E_o$ (defined as the net radiation expressed as an equivalent depth of liquid
water, Budyko, 1974). We then calculate the aridity index ($\overline{E_o}/\overline{P}$) using $P$ from the CDR and $E_o$ from the SRB
databases (see Fig. S1a in the Supplementary Material).

On general grounds, we anticipate that two important factors likely to control the partitioning of hydrologic
variability were the water storage capacity and the presence of ice/snow at the surface. For the storage, we estimate
the water storage capacity ($S_{max}$) using the monthly $\Delta S$ data in CDR database. The water storage $S(t)$ at each time





step $t$ (monthly here) was first calculated from the accumulation of $\Delta S(t)$, i.e., $S(t) = S(t-1) + \Delta S(t)$ where we

assumed zero storage at the beginning of the study period (i.e., $S(0) = 0$). With the resulting time series available,

$S_{\max}$ was estimated as the difference between the maximum and minimum $S(t)$ during the study period at each

grid-box (see Fig. S1b in the Supplementary Material). The estimated $S_{\max}$ shows a large range from 0 to 1000

mm with the majority of values from 50 to 600 mm (Fig. S1b), which generally agrees with global rooting depth

estimates assuming that water occupies from 10 to 30% of the soil volume at field capacity (Jackson et al., 1996;

Wang-Erlandsson et al., 2016; Yang et al., 2016). To characterise snow/ice cover, and to distinguish extremely

hot and cold regions, we also make use of a gridded global land air temperature dataset from the Climatic Research

Unit (CRU TS4.01 database, monthly, 1901-2016, $0.5° \times 0.5°$) (Harris et al., 2014). (see Fig. S1c in the

Supplementary Material).

2.3 Spatial Mask to Define Study Extent

The CDR database provides an estimate of the uncertainty ($\pm 1\sigma$) for each of the hydrologic variables ($P$, $E$, $Q$,

$\Delta S$) in each month. We use those uncertainty estimates to identify and remove regions with high relative

uncertainty in the CDR data. The relative uncertainty is calculated as the ratio of root mean square of the

uncertainty ($\pm 1\sigma$) to the mean annual $P$, $E$ and $Q$ at each grid-box following the procedure used by Milly and

Dunne (2002a). Note that the long term mean $\Delta S$ is zero by construction in the CDR database, and for that reason

we did not use $\Delta S$ to calculate the relative uncertainty. Grid-boxes with a relative uncertainty (in $P$, $E$ and $Q$) more

than 0.1 are deemed to have high relative uncertainty (Milly and Dunne, 2002a) and were excluded from the study

extent. The excluded grid-boxes were mostly in the Himalayan region, the Sahara Desert and in Greenland. The

final spatial mask is shown in Fig. 1 and this has been applied throughout this study.

2.4 Further Evaluation of CDR Data for Variability Analysis

In the original work, the CDR database was validated by comparison with independent observations including (i)

mean seasonal cycle of $Q$ from 26 large basins (see Fig. 8 in Zhang et al., 2018), (ii) mean seasonal cycle of $\Delta S$

from 12 large basins (Fig. 10 in Zhang et al., 2018), (iii) monthly runoff from 165 medium size basins and a

further 862 small basins (Fig. 14 in Zhang et al., 2018), (iv) summer $E$ from 47 flux towers (Fig. 16 in Zhang et

al., 2018). Those evaluations did not directly address variability in various water cycle elements. With our focus





on the variability we decided to conduct further validations of the CDR database beyond those described in the
original work. In particular, we focussed on further independent assessments of $E$ and we use monthly (as opposed
to summer) observations of $E$ from FLUXNET to evaluate the variability in $E$. We also compare the CDR with
two other gridded global $E$ products that were not used to develop the CDR including LandFluxEval ($1° \times 1°$,
monthly, 1989-2005) (Mueller et al., 2013) and the Max Planck Institute (MPI, $0.5° \times 0.5°$, monthly, 1982-2011)
(Jung et al., 2010) product.

For the comparison to FLUXNET observations (Baldocchi et al., 2001; Agarwal et al., 2010) we identified 32
flux tower sites (site locations are shown in Fig. S2 and details are shown in Table S1) having at least three years
of continuous (monthly) measurements using the FluxnetLSM R package (v1.0) (Ukkola et al. 2017). The monthly
totals and annual climatology of $P$ and $E$ from CDR generally follow FLUXNET observations, with high
correlations and reasonable Root Mean Square Error (Figs. S3-S4, Table S1). Comparison of the point-based
FLUXNET ($\sim$ 100 m – 1 km scale) with the grid-based CDR ($\sim$ 50 km scale) is problematic since the CDR
represents an area that is at least 2500 times larger than the area represented by the individual FLUXNET towers
and we anticipate that the CDR record would be "smoothed" relative to the FLUXNET record. With that in mind,
we chose to compare the ratio of the standard deviation of $E$ to $P$ between the CDR and FLUXNET databases and
this normalised comparison of the hydrologic variability proved encouraging (Fig. S5).

As a further evaluation, we compare gridded $E$ data in the CDR database against two other global $E$ databases
including LandFluxEVAL ($1° \times 1°$, monthly, 1989-2005) (Mueller et al., 2013) and Max Planck Institute (MPI,
$0.5° \times 0.5°$, monthly, 1982-2011) (Jung et al., 2010) that were not used to construct the CDR database. We found
that monthly mean $E$ from the CDR database is slightly underestimated compared with LandFluxEVAL database
(Fig. S6a), but agrees closely with the MPI database (Fig. S7a). In terms of variability, the standard deviations of
monthly $E$ from the CDR are slightly different than those in the MPI database (Fig. S7c) but were in very close
agreement with the LandFluxEVAL database (Fig. S6c).

In summary, we concluded that the CDR database was suitable for an initial investigation of the inter-annual
variability in the water cycle.

**3. Mean and Variability of Water Cycle Components**



3.1 Mean Annual *P*, *E*, *Q* and the Budyko Curve

The global pattern of mean annual *P*, *E*, *Q* using the CDR data (1984-2007) is shown in Fig. 2. The mean annual
$P$ ($\bar{P}$) is prominent in tropical regions, southern China, eastern and western North America (Fig. 2a). The
magnitude of mean annual $E$ ($\bar{E}$) more or less follows the pattern of $\bar{P}$ in the tropics (Fig. 2b) while the mean
annual $Q$ ($\bar{Q}$) is particularly prominent in the Amazon, South and Southeast Asia, tropical parts of west Africa
and in some other coastal regions at higher latitudes (Fig. 2c).

We relate the grid-box level ratio of $\bar{E}$ to $\bar{P}$ in the CDR database to the classical Budyko (1974) curve using the
aridity index ($\overline{E_o}/\bar{P}$) (Fig. 3a). As noted previously, in the CDR database, $\overline{\Delta S}$ is forced to be zero and this enforced
steady state allowed us to also predict the ratio of $\bar{Q}$ to $\bar{P}$ using the same Budyko curve (Fig. 3b). The Budyko
curves follow the overall trend in the CDR data. However, there is substantial scatter due to, for example, regional
variations related to seasonality, water storage change and physics of runoff generation (Milly, 1994a, b). The
overall patterns are as expected with $\bar{E}$ following $\bar{P}$ in dry environments ($\overline{E_o}/\bar{P} > 1.0$) while $\bar{E}$ follows $\overline{E_o}$ in wet
environments ($\overline{E_o}/\bar{P} \leq 1.0$) (Fig. 3).

3.2 Inter-annual Variability in *P*, *E*, *Q* and $\Delta S$

We use the variance balance equation (Eq. 2) to partition the inter-annual $\sigma_P^2$ into separate components due to $\sigma_E^2$,
$\sigma_Q^2$, $\sigma_{\Delta S}^2$ along with the three covariance components ($2cov(E,Q)$, $2cov(E,\Delta S)$, $2cov(Q,\Delta S)$) (Fig. 4). The
spatial pattern of $\sigma_P^2$ (Fig. 4a) is very similar to that of $\bar{P}$ (Fig. 2a), which implies that the $\sigma_P^2$ is positively
correlated with $\bar{P}$. In contrast the partitioning of $\sigma_P^2$ to the various components is very different from the
partitioning of $\bar{P}$ (cf. Fig. 2 and 4). First we note that while the overall spatial pattern of $\sigma_E^2$ more or less follows
$\sigma_P^2$, the overall magnitude of $\sigma_E^2$ is much smaller than $\sigma_P^2$ and $\sigma_Q^2$ in most regions, and in fact $\sigma_E^2$ is also generally
smaller than $\sigma_{\Delta S}^2$. The prominence of $\sigma_{\Delta S}^2$ (compared to $\sigma_E^2$) surprised us. The three covariance components
($cov(E,Q)$, $cov(E,\Delta S)$, $cov(Q,\Delta S)$) are also important in some regions. In more detail, the $cov(E,Q)$ term is
prominent in regions where $\sigma_Q^2$ is large and is mostly negative in those regions (Fig. 4e), indicating that years with
lower *E* are associated with higher *Q* and vice-versa. There are also a few regions with prominent positive values
for $cov(E,Q)$ (e.g., the seasonal hydroclimates of northern Australia) indicating that in those regions, years with
a higher *E* are associated with higher *Q*. The $cov(E,\Delta S)$ term (Fig. 4f) has a similar spatial pattern to the





$cov(E, Q)$ term (Fig. 4e) but with a smaller overall magnitude. Finally, the $cov(Q, \Delta S)$ term shows a more
complex spatial pattern, with both prominent positive and negative values (Fig. 4g) in regions where $\sigma_Q^2$ (Fig. 4c)
and $\sigma_{\Delta S}^2$ (Fig. 4d) are both large.

These results show that the spatial patterns in variability are not simply a reflection of patterns in the long-term
mean state. On the contrary, we find that of the three primary variance terms, the overall magnitude of (inter-
annual) $\sigma_E^2$ is the smallest implying the least (inter-annual) variability in $E$. This is very different from the
conclusions based on spatial patterns in the mean $P$, $E$ and $Q$ (see previous section). Further, while $\sigma_Q^2$ more or
less follows $\sigma_P^2$ as expected, we were surprised by the magnitude of $\sigma_{\Delta S}^2$ which, in general, substantially exceeds
the magnitude of $\sigma_E^2$. Further, the magnitude of the covariance terms can be important, especially in regions with
high $\sigma_Q^2$. However, unlike the variances, the covariance can be both positive and negative and this introduces
additional complexity. For example, with a negative covariance it is possible for the variance in $Q$ ($\sigma_Q^2$) to exceed
the variance in $P$ ($\sigma_P^2$). To examine that in more detail we calculated the equivalent frequency distribution for each
of the plots in Fig. 4. The results (Fig. 5) further emphasise that in general, $\sigma_E^2$ is the smallest of the variances (Fig.
5b). We also note that the frequency distributions for the covariances (Fig. 5efg) are not symmetrical. In summary,
it is clear that spatial patterns in the inter-annual variability of the water cycle (Fig. 4) do not simply follow the
spatial patterns for the inter-annual mean (Fig. 2).

3.3 Relation Between Variability and the Mean State for $P$, $E$, $Q$

Differences in the spatial patterns of the mean (Fig. 2) and inter-annual variability (Fig. 4) in the global water
cycle led us to further investigate the relation between the mean and the variability for each separate component.
Here we relate the standard deviation ($\sigma_P$, $\sigma_E$, $\sigma_Q$) instead of the variance to the mean of each water balance flux
(Fig. 6) since the standard deviation has the same physical units as the mean making the results more comparable.
As inferred previously, we find $\sigma_P$ to be positively correlated with $\bar{P}$ but with substantial scatter (Fig. 6a). The
same result more or less holds for the relation between $\sigma_Q$ and $\bar{Q}$ (Fig. 6c). In contrast the relation between $\sigma_E$ and
$\bar{E}$ is very different (Fig. 6b). In particular, $\sigma_E$ is a small fraction of $\bar{E}$ and this complements the earlier finding (Fig.
6b) that the inter-annual variability for $E$ is generally smaller than for the other physical variables, $P$, $Q$ or $\Delta S$.
(The same result was also found using both LandFluxEVAL and MPI databases, see Fig. S8 in the Supplementary
Material.) Importantly, unlike $P$ and $Q$, $E$ is constrained by both water and energy availability (Budyko, 1974)




and the limited inter-annual variability in $E$ presumably reflects limited inter-annual variability in the available
(radiant) energy ($E_o$). This is something that could be investigated in a future study.

**4. Relating the Variability of $P$ $E$, $Q$ and $\Delta S$ to Aridity**

In the previous section, we investigated spatial patterns of the mean and the variability in the global water cycle.
In this section, we extend that by investigating the partitioning of $\sigma_P^2$ to the three primary physical terms ($\sigma_E^2$, $\sigma_Q^2$,
$\sigma_{\Delta S}^2$) along with the three relevant covariances. For that, we begin by comparing the Koster and Suarez (1999)
theory against the CDR data and then investigate how the partitioning of the variance is related to the aridity index
$\overline{E_o}/\overline{P}$ (see Fig. S1a in the Supplementary Material). Following that, we investigate variance partitioning in relation
to both our estimate of the storage capacity $S_{max}$ (see Fig. S1b in the Supplementary Material) as well as the mean
annual air temperature $\overline{T_a}$ (see Fig. S1c in the Supplementary Material) that we use as a surrogate for snow/ice
cover. We finalise this section by examining the partitioning of variance at three selected study sites that represent
extremely dry/wet, high/low water storage capacity and the hot/cold spectrums.

4.1 Comparison with the Koster and Suarez (1999) Theory

We first evaluate the classical empirical curve of Koster and Suarez (1999) by relating ratios $\sigma_E/\sigma_P$ and $\sigma_E/\sigma_P$ to
the aridity index (Fig. 7). The ratio $\sigma_E/\sigma_P$ in the CDR database is generally overestimated by the empirical Koster
and Suarez curve, especially in dry environments (e.g., $\overline{E_o}/\overline{P} > 3$). The inference here is that the Koster and Suarez
theory predicts $\sigma_E/\sigma_P$ to approach unity in dry environments while the equivalent value in the CDR data is
occasionally unity but is generally smaller. With $\sigma_E/\sigma_P$ generally overestimated by the Koster and Suarez theory
we expect, and find, that $\sigma_Q/\sigma_P$ is underestimated by the same theory (Fig. 7b). The same overestimation was
found based on the other two independent databases for $E$ (LandFluxEVAL and MPI) (Fig. S9). This
overestimation is discussed further in section 5.

4.2 Relating Inter-annual Variability to Aridity

Here we examine how the fraction of the total variance in precipitation accounted for by the three primary variance
terms along with the three covariance terms varies with the aridity index ($\overline{E_o}/\overline{P}$) (Fig. 8). (Also see Fig. S10 for





the spatial maps.) The ratio $\sigma_E^2/\sigma_P^2$ is close to zero in extremely wet regions and has an upper limit noted
previously (Fig. 7a) that approaches unity in extremely dry regions (Fig. 8a). The ratio $\sigma_Q^2/\sigma_P^2$ is close to zero in
extremely dry regions but approaches unity in extremely wet regions but with substantial scatter (Fig. 8b). The
ratio $\sigma_{\Delta S}^2/\sigma_P^2$ is close to zero in both extremely dry/wet regions (Fig. 8c) but shows the largest range at an
intermediate aridity index ($\overline{E_o}/\overline{P} \sim 1.0$).

The covariance ratios are all small in extremely dry (e.g., $\overline{E_o}/\overline{P} \geq 6.0$) environments and generally show the largest
range in semi-arid and humid environments. The peak magnitudes for the three covariance components
consistently occur when $\overline{E_o}/\overline{P}$ is close to 1.0 which is the threshold often used to separate wet and dry
environments.

4.3 Further Investigations on the Factors Controlling Partitioning of the Variance

The previous results (Sections 4.1 and 4.2) have demonstrated that spatial variation in the partitioning of $\sigma_P^2$ into
$\sigma_E^2$, $\sigma_Q^2$, $\sigma_{\Delta S}^2$ and the three covariance components is complex. To help further understand inter-annual variability
of the terrestrial water cycle, we conduct further investigations in this section using two factors likely to have a
major influence on the variance partitioning of $\sigma_P^2$. The first is the storage capacity $S_{max}$ (see Fig. S1b in the
Supplementary Material). The second is the mean annual air temperature $\overline{T_a}$ (see Fig. S1c in the Supplementary
Material) which is used here as a surrogate for snow/ice presence.

4.3.1 Relating Inter-annual Variability to Storage Capacity

We first relate the partitioning of $\sigma_P^2$ to water storage capacity ($S_{max}$) by repeating Fig. 8 but instead we use a
logarithmic scale for the x-axis and we distinguish $S_{max}$ via the background colour (Fig. 9). To eliminate the
possible overlap of grid-cells in the colouring process, all the grid-cells over land are further separated using
different latitude ranges (as shown in the four columns of Fig. 9), i.e., 90N-60N, 60N-30N, 30N-0 and 0-90S. We
find that $S_{max}$ is relatively high in wet environments ($\overline{E_o}/\overline{P} \leq 1.0$) but shows no obvious relation with the
partitioning of $\sigma_P^2$. However, in dry environments ($\overline{E_o}/\overline{P} > 1.0$) the ratio $\sigma_E^2/\sigma_P^2$ apparently decreases with the
increase of $S_{max}$ (Fig. 9a-d). That relation is particularly obvious in extremely dry environments ($\overline{E_o}/\overline{P} \geq 6.0$) at
equatorial latitudes where there is an upper limit of $\sigma_E^2/\sigma_P^2$ close to 1.0 when $S_{max}$ is small (blue grid-cells in Fig.



9c). The interpretation for those extremely dry environments is that when $S_{max}$ is small, $\sigma_P^2$ is almost completely
partitioned into $\sigma_E^2$ (Fig. 9bc) with the other variance and covariance components close to zero. While for those
same extremely dry environments, as $S_{max}$ increases, the partitioning of $\sigma_P^2$ is shared between $\sigma_E^2$ and $\sigma_{\Delta S}^2$ and their
covariance (Fig. 9cks) with $\sigma_Q^2$ and its covariance components close to zero (Fig. 9gow). However, at polar
latitudes in the northern hemisphere (panels in the first and second columns of Fig. 9) there are variations that
could not be easily associated with variations in $S_{max}$ which led us to investigate the role of snow/ice on the
variance partitioning in the following section.

4.3.2 Relating Inter-annual Variability to Mean Air Temperature

To understand the potential role of snow/ice in modifying the variance partitioning, we repeat the previous
analysis (Fig. 9) but here we use the mean annual air temperature ($\overline{T_a}$) to colour the grid-cells to crudely identify
the presence of snow/ice (Fig. 10). Most of the variations at polar latitudes in the northern hemisphere (panels in
the first and second columns of Fig. 10) is associated with low air temperature (e.g., $\overline{T_a} < 0$ °C in blue colour),
making the results associated with high air temperature (e.g., $\overline{T_a} > 10$ °C in green-yellow-red colours) relatively
more compact. That pattern is particularly obvious in extremely wet environment, where the ratio $\sigma_Q^2/\sigma_P^2$ is close
to 1.0 when $\overline{T_a}$ is high (e.g., $\overline{E_o}/\overline{P} \leq 0.5$ and $\overline{T_a} > 10$ °C, with green-yellow-red grid-cells on the panels in the
second row of Fig. 10) with the other variance-covariance components close to zero. This indicates that in
extremely wet environment, when $\overline{T_a}$ is high, $\sigma_P^2$ is almost completely partitioned into $\sigma_Q^2$. However, when $\overline{T_a}$ is
low in extremely wet environment, there are substantial variations in all variance-covariance components
(e.g., $\overline{E_o}/\overline{P} \leq 0.5$ and $\overline{T_a} < 0$ °C, see the blue grid-cells on the panels in the first column of Fig. 10). That result
indicates the complexity of variance partitioning associated with the presence of snow/ice.

4.4 Case Studies

The previous results (Section 4.3) have demonstrated that the partitioning of $\sigma_P^2$ is predominantly influenced by
the water storage capacity ($S_{max}$) in extremely dry environments ($\overline{E_o}/\overline{P} \geq 6.0$) and by mean air temperature ($\overline{T_a}$)
in extremely wet environments ($\overline{E_o}/\overline{P} \leq 0.5$). In this section, we examine, in greater detail, several sites to gain
deeper understanding of the partitioning of $\sigma_P^2$. For that purpose, we selected three sites based on extreme values
for the three explanatory parameters, i.e., $\overline{E_o}/\overline{P}$ (Fig. S1a), $S_{max}$ (Fig. S1b) and $\overline{T_a}$ (Fig. S1c). The criteria to select





three climate sites are as follows, Site 1: dry ($\overline{E_o}/\overline{P} \geq 6.0$) and small $S_{max}$ ($S_{max} \approx 0$), Site 2: dry ($\overline{E_o}/\overline{P} \geq 6.0$) and
relatively large $S_{max}$ ($S_{max} \gg 0$) and Site 3: wet ($\overline{E_o}/\overline{P} \leq 0.5$) and hot ($\overline{T_a} > 25$ °C). For each of the three sites, we
use a representative grid-cell (Fig. 11) to show the original time series (Fig. 12) and the partitioning of variability
(Fig. 13).

We show the $P$, $E$, $Q$ and $\Delta S$ time series along with the relevant variances and covariances in Fig. 12. Starting
with the two dry sites, at the site with low storage capacity (Site 1), the time series shows that $E$ closely follows
$P$ leaving annual $Q$ and $\Delta S$ close to zero (Fig. 12a). The variance of $P$ ($\sigma_P^2 = 206.9$ mm$^2$) is small and almost
completely partitioned into the variance of $E$ ($\sigma_E^2 = 196.9$ mm$^2$), leaving very limited variance for $Q$, $\Delta S$ and all
three covariance components (Fig. 12b). At the site with high storage capacity (Site 2), $E$, $Q$ and $\Delta S$ do not simply
follow $P$ (Fig. 12c). As a consequence, the variance of $P$ ($\sigma_P^2 = 2798.0$ mm$^2$) is shared between $E$ ($\sigma_E^2 = 1150.2$
mm$^2$), $\Delta S$ ($\sigma_{\Delta S}^2 = 800.5$ mm$^2$) and their covariance component ($2cov(E, \Delta S) = 538.4$ mm$^2$, Fig. 12d). Switching
now to the remaining wet and hot site (Site 3), $Q$ closely follows $P$, with $\Delta S$ close to zero and $E$ showing little
inter-annual variation (Fig. 12e). The variance of $P$ ($\sigma_P^2 = 57374.4$ mm$^2$) is relatively large and almost completely
partitioned into the variance of $Q$ ($\sigma_Q^2 = 57296.4$ mm$^2$), leaving very limited variance for $E$ and $\Delta S$ and the three
covariance components (Fig. 12f). We also examined numerous other sites with similar extreme conditions as the
three case study sites and found the same basic patterns as reported above.

To put the data from the three case study sites into a broader variability context we position the site data onto a
backdrop of original Fig. 8. As noted previously, at Site 1, the ratio $\sigma_E^2/\sigma_P^2$ is very close to unity (Fig. 13a), and
under this extreme condition, we have the following approximation,
$$\sigma_P^2 \approx \sigma_E^2 \quad \text{(Site 1, dry and } S_{max} \approx 0) \tag{3}$$
In contrast, for Site 2 with the same aridity index but higher $S_{max}$, we have,
$$\sigma_P^2 \approx \sigma_E^2 + \sigma_{\Delta S}^2 + 2cov(E, \Delta S) \quad \text{(Site 2, dry and } S_{max} \gg 0) \tag{4}$$
Finally, at Site 3, we have,
$$\sigma_P^2 \approx \sigma_Q^2 \quad \text{(Site 3, wet and hot)} \tag{5}$$

4.5 Synthesis





The above simple examples demonstrate that aridity $\overline{E_o}/\overline{P}$, storage capacity $S_{max}$ and air temperature $\overline{T_a}$ all play
roles in the partitioning of $\sigma_P^2$ to the various components. Our synthesis of the results for the partitioning of $\sigma_P^2$ is
summarised in Fig. 14. In dry and $S_{max} \approx 0$ environments we have minimal runoff and expect that $\sigma_P^2$ is more or
less completely partitioned into $\sigma_E^2$ (Fig. 14a). In those environments, (inter-annual) variations in storage $\sigma_{\Delta S}^2$ play
a limited role in setting the overall variability. However, in dry and $S_{max} \gg 0$ environments, $\sigma_E^2$ is only a fraction
of $\sigma_P^2$ leaving the overall variance attributed to $\sigma_{\Delta S}^2$ and the covariance between $E$ and $\Delta S$ (Fig. 14c and Fig. 14e).
This implies the hydrological importance of water storage capacity in buffering variations of the water cycle under
dry conditions.

Under extremely wet conditions, the huge difference in variance partitioning occurs between the hot and cold
conditions instead of water storage capacity conditions in dry conditions. In wet and hot environments, we have
maximum runoff and expect that $\sigma_P^2$ is more or less completely partitioned into $\sigma_Q^2$ (Fig. 14b), and the variations
in evapotranspiration $\sigma_E^2$ and storage $\sigma_{\Delta S}^2$ play a limited role in setting the overall variability. However, in wet and
cold environments, the variance partitioning shows great complexity, with $\sigma_Q^2/\sigma_P^2$ and $\sigma_{\Delta S}^2/\sigma_P^2$ vary a lot caused
by snow/ice melting. This signifies the hydrological importance of thermal processes (melting/freezing) under
extremely cold conditions.

The most complex patterns to interpret are those for semi-arid to semi-humid environments (i.e., $\overline{E_o}/\overline{P} \sim 1.0$). In
those environments, the three covariance terms all play important roles and we found that simple environmental
gradients (e.g., dry/wet, high/low storage capacity, hot/cold) could not easily explain the observed patterns. A
major effort will be needed to discover the controlling factors for variability of the water cycle in these
environments.

**5. Discussion**

Importantly, hydrologists have long been aware that the water storage effects were going to be important for
understanding water cycle variability (e.g., Milly and Dunne, 2002b; Zhang et al., 2008; Donohue et al., 2010;
Wang and Alimohammadi, 2012), but without readily available databases it has been difficult to quantify water
cycle variability in a consistent way. For example, we are not aware of maps showing global spatial patterns in
variance for any terms of the water balance (except for $P$). In this study, we have used a recently released global



gridded hydrologic re-analysis product, i.e., the Climate Data Record (CDR) to conduct an initial investigation of
inter-annual variability in the terrestrial branch of the global water cycle. To the best of our knowledge, the results
in our manuscript present the first attempt to gain a global overview of the magnitude for various terms (Eq. 2)
that document variability in the water cycle. Our results demonstrate that the global patterns of inter-annual
variability in the water cycle do not simply follow those of the long-term mean. In particular, with the variance
calculations, the annual anomalies are squared and hence do not cancel out (like they do when calculating the
mean). Hence we were initially surprised that the inter-annual variability of water storage change ($\sigma_{\Delta S}^2$) is typically
larger than the inter-annual variability of evapotranspiration ($\sigma_E^2$). Moreover, the covariance components are also
prominent and can be negative, which means that it is possible for the variability in the sinks (e.g., $\sigma_Q^2$, $\sigma_{\Delta S}^2$) can
actually exceed the variability in the source ($\sigma_P^2$) (Eq. 2).

Our further analysis based on six climate end members, dry/wet, high/low water storage capacity and hot/cold
offered some further general insights about hydrologic variability. For example, under extremely dry (water-
limited) conditions, with limited storage capacity ($S_{max}$) we found that $E$ follows $P$ and $\sigma_E^2$ follows $\sigma_P^2$, with $\sigma_Q^2$
and $\sigma_{\Delta S}^2$ approaching zero. However, as $S_{max}$ increases, the partitioning of $\sigma_P^2$ progressively shifts to a balance
between $\sigma_E^2$, $\sigma_{\Delta S}^2$ and cov ($E$, $\Delta S$) (Fig. 12-14). Under extremely wet (energy-limited) and hot environments (i.e.,
no snow/ice impact) we found the inter-annual variations in $P$ mostly be partitioned to inter-annual variations in
$Q$ (with both $\sigma_E^2$ and $\sigma_{\Delta S}^2$ approaching zero). However, in wet environments that were cold, we expected thermal
processes (freeze/melt) to play a critical role in the hydrologic variability. Our results confirm that, with the
finding that hydrologic partitioning of variability was highly (spatially) variable under extremely cold conditions
(Fig. 12-14) and we were unable to provide any useful simplifications to summarise the data. These results
highlight a key point that while the long-term mean state is not especially sensitive to variations in hydrologic
water storage or phase, the long-term variability is very sensitive to those same variations.

The most complex results were found in semi-arid/semi-humid ($0.5 < \overline{E_o}/\overline{P} < 1.5$) environments, where all three
covariances (Eq. 2) were found to play critical roles in the overall partitioning of variability (Figs. 4-5). In many
regions, the (absolute) magnitudes of the covariances were actually larger than the variances of the water balance
components $E$, $Q$ and $\Delta S$ (e.g., Fig. 8). That result demonstrates that deeper understanding of the process-level
interactions that are embedded within each of the three covariance terms is still needed to help understand
variability in the water cycle in these biologically productive regions ($0.5 < \overline{E_o}/\overline{P} < 1.5$).






This study should be viewed as an initial investigation of the inter-annual variability in the global land water cycle.
We managed to obtain some syntheses based on the availability of current data, and we expect that with the
improvement of hydrologic databases over the coming years some of the detailed spatial patterns may change.
However, even from this initial investigation, some general principles do already appear clear. One general finding
is that the global pattern in the partitioning of inter-annual variability in the water cycle is not simply a reflection
of patterns in the partitioning of the long-term mean. For example, while the inter-annual water storage change is
often (safely) assumed to be negligible in terms of the long-term mean state, it is clear that storage variations are
central to understanding inter-annual variability of global water cycle. A second generalisation is that the
covariance components (Eq. 2) can be relatively large and are negative in some regions. The consequence is that
variability in the sinks (e.g., $\sigma_Q^2$, $\sigma_{\Delta S}^2$) can, and do, exceed the variability in the source ($\sigma_P^2$), especially in
biologically productive regions (Fig. 4).

The syntheses of the long-term mean water cycle originated in 1970s (Budyko, 1974), and it took several decades
for those general principles to become widely adopted in the hydrologic community. It remains a challenge to
develop a synthesis of hydro-climatic variability in the terrestrial branch of the water cycle, and major intellectual
efforts will be needed to develop generally applicable principles.

**6. Conclusions**

In this study, we describe an initial investigation of the inter-annual variability of the terrestrial branch in the
global water cycle that uses the recently released global monthly Climate Data Record (CDR) database for $P$, $E$,
$Q$ and $\Delta S$. We start by investigating the partitioning of $P$ in the water cycle in terms of long-term mean and then
extend that to the inter-annual variability. While the mean annual $P$ is mostly partitioned into mean annual $E$ and
$Q$, as is well known. However, we find that the variance of $P$ ($\sigma_P^2$) is mostly partitioned into the variance of $Q$ ($\sigma_Q^2$)
and variance of $\Delta S$ ($\sigma_{\Delta S}^2$). This result indicates that the global patterns of inter-annual variability in the water cycle
do not simply follow the long-term mean. A second general finding is that the covariance components are
important and can be negative in some regions, indicating the variability in the sinks (e.g., $\sigma_Q^2$, $\sigma_{\Delta S}^2$) can, and do,
exceed the variability in the source ($\sigma_P^2$). Our attempts to develop deeper understanding of variance partitioning
led to some syntheses in extreme environments (wet/dry vs hot/cold). In particular, we find that in extremely dry




environments (either hot/cold) the partitioning of $\sigma_P^2$ is closely related to the water storage capacity. With limited
storage capacity, the partitioning of $\sigma_P^2$ is mostly to $\sigma_E^2$ but as the storage capacity increases, the partitioning of
$\sigma_P^2$ is increasingly shared between $\sigma_E^2$ and $\sigma_{\Delta S}^2$ and the covariance between those variables (Fig. 14). In contrast,
in extremely wet environments, there are large divergences in the variance partitioning between hot and cold
conditions. In hot conditions, $\sigma_P^2$ is mostly partitioned to $\sigma_Q^2$ but under cold conditions, $\sigma_P^2$ is partitioned to all
available variability sinks (Fig. 14). However, in biologically productive semi-arid/semi-humid ($0.5 < \overline{E_o}/\overline{P} < 1.5$)
environments, we found the variance partitioning to be very complex and that partitioning was not obviously
associated with simple environmental factors. A general understanding of hydro-climatic variability remains a
major intellectual challenge and we anticipate major efforts will be needed to synthesise general principles that
cover the full spectrum of hydrologic variability.

**Acknowledgements**
This research was supported by the Australian Research Council (CE11E0098, CE170100023), and D.Y.
acknowledges support by the National Natural Science Foundation of China (51609122). The authors declare that
there is no conflict of interests regarding the publication of this paper. All data used in this paper are available
online as referenced in the 'Methods and Data' section.

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





**List of Figures:**




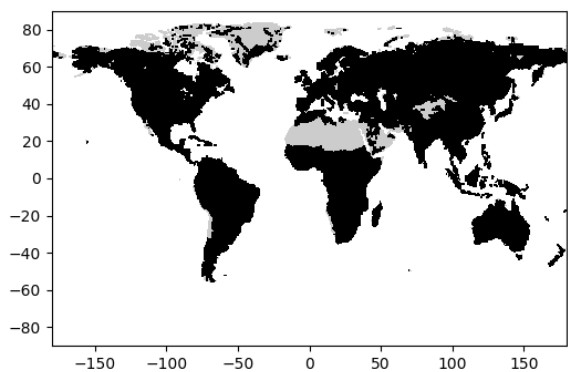


**Figure 1. Spatial mask used in this study. Grey areas (Himalayan region, Sahara Desert, Greenland) have been**

**masked out of the CDR database.**






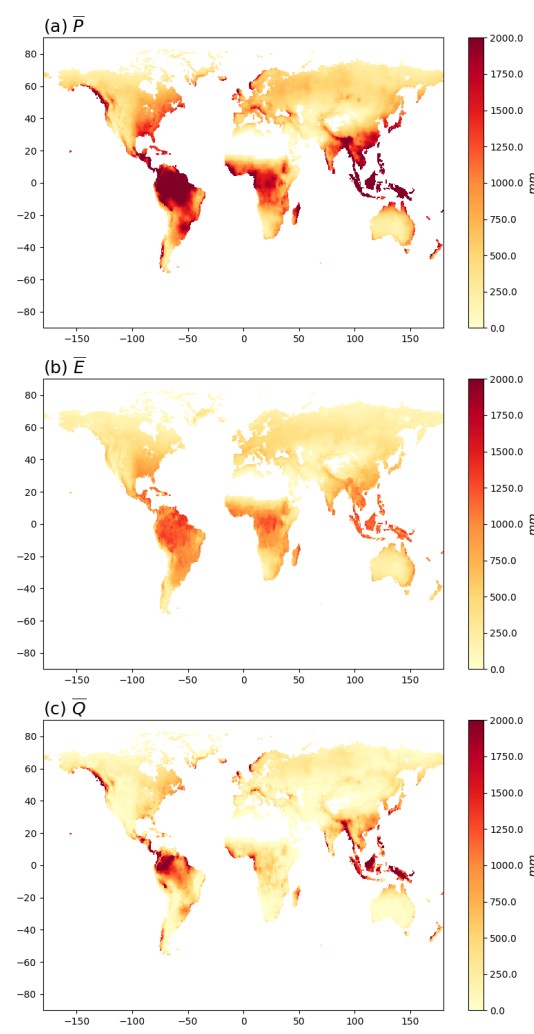


**Figure 2. Mean annual (1984-2010) (a) *P*, (b) *E* and (c) *Q*. Note that the mean annual Δ*S* in the CDR database is zero**
**by construction and is not shown.**




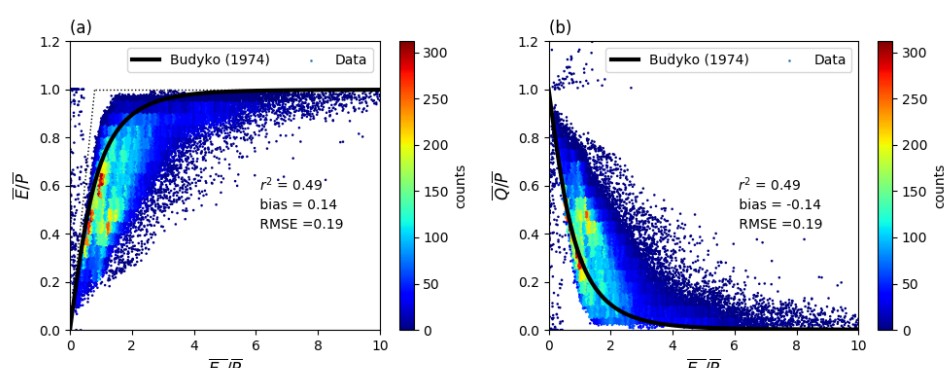


**Figure 3. Relationship of mean annual (a) evapotranspiration ($\overline{E}/\overline{P}$) and (b) runoff ($\overline{Q}/\overline{P}$) ratios to the aridity index**

**($\overline{E_o}/\overline{P}$) from the CDR and SRB databases. For comparison, the Budyko (1974) curve is shown on the left panel (Fig.**

**3a). The curve on the right panel (Fig. 3b) is calculated assuming a steady state ($\overline{Q}/\overline{P} = 1 - \overline{E}/\overline{P}$).**







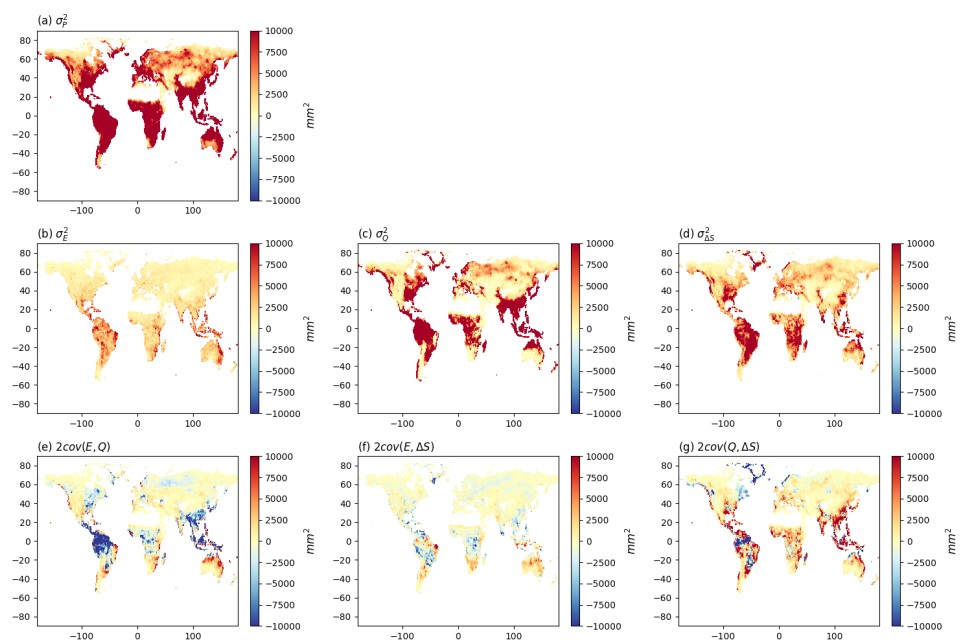


**Figure 4.** Water cycle variances ($\sigma_P^2$, $\sigma_E^2$, $\sigma_Q^2$, $\sigma_{\Delta S}^2$) and covariances ($cov(E, Q)$, $cov(E, \Delta S)$, $cov(Q, \Delta S)$). Note that we

have multiplied the covariances by two (see Eq. 2).





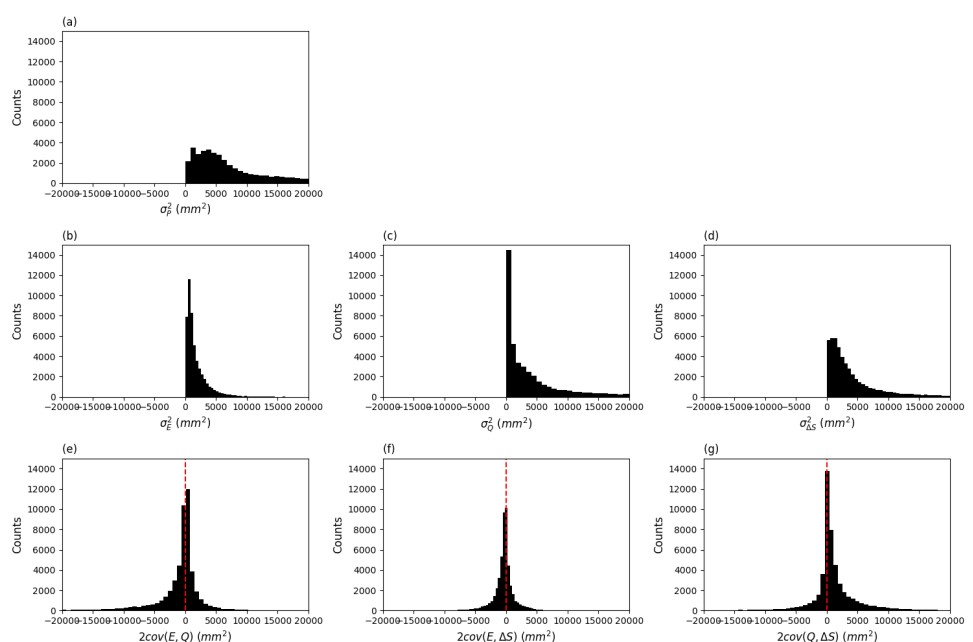


**Figure 5. Distribution of water cycle variances ($\sigma_P^2$, $\sigma_E^2$, $\sigma_Q^2$, $\sigma_{\Delta S}^2$) and covariances ($cov(E, Q)$, $cov(E, \Delta S)$, $cov(Q, \Delta S)$).**

**Note that we have multiplied the covariances by two (see Eq. 2).**







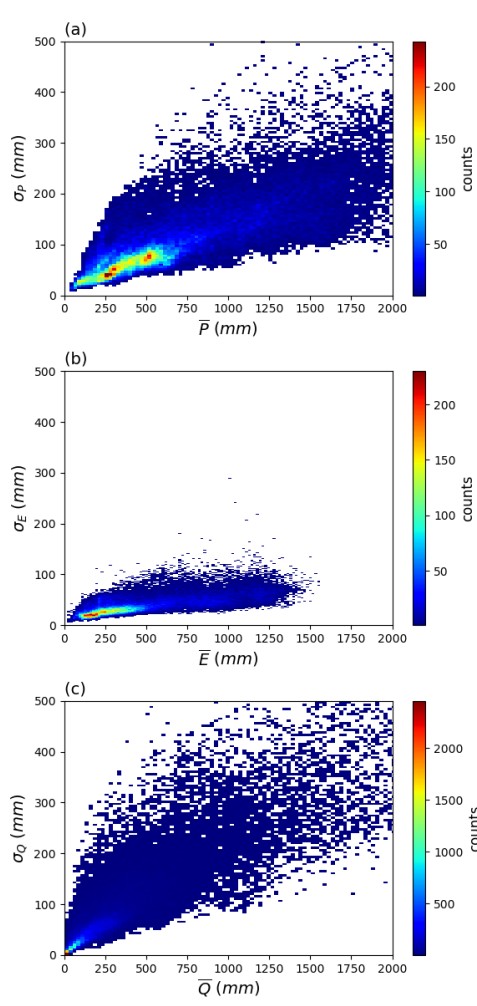


**Figure 6. Relation between inter-annual mean and standard deviation for (a) *P*, (b) *E* and (c) *Q* from the CDR**

**database. Note that the mean annual Δ*S* is zero by construction and is not shown.**







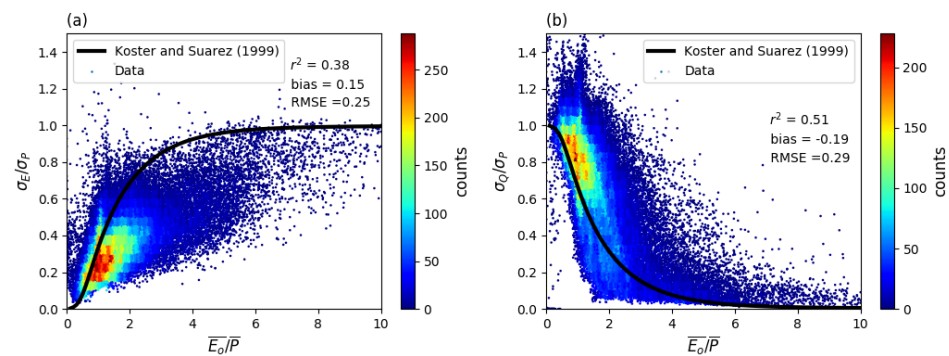


**Figure 7. Relationship of inter-annual standard deviation of (a) evapotranspiration ($\sigma_E/\sigma_P$) and (b) runoff ($\sigma_Q/\sigma_P$)**


**ratios to aridity ($\overline{E_o}/\overline{P}$). The curves represent the semi-empirical relations from Koster and Suarez (1999).**








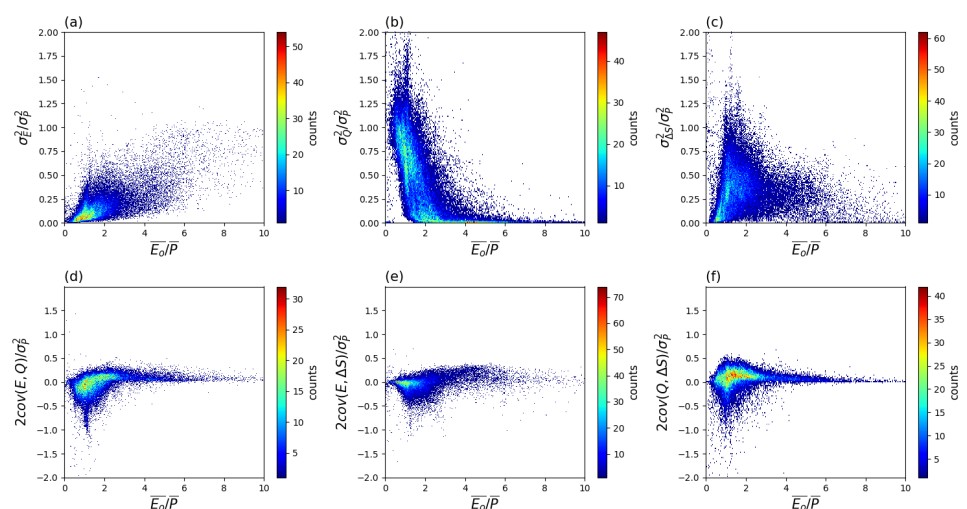


**Figure 8. Relation between water cycle variances-covariances (see Fig. 4b-g) as a fraction of the variance of $P$ ($\sigma_P^2$) and the aridity index ($\overline{E_o}/\overline{P}$) coloured by density. Note that we have multiplied the covariance components by two (see Eq. 2).**





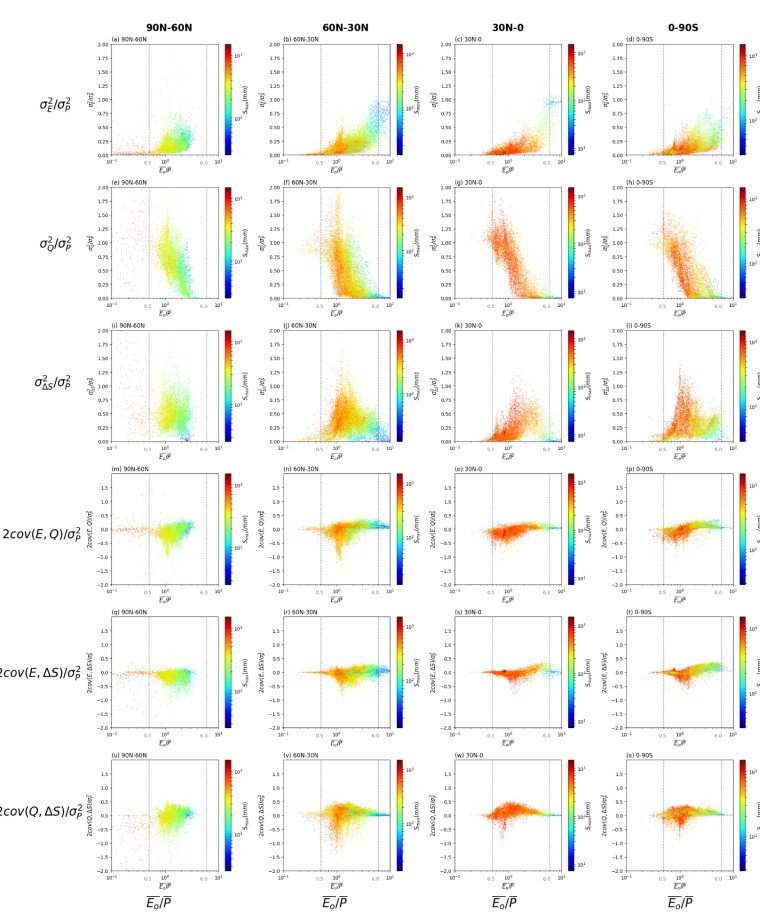


**Figure 9. Relation between water cycle variances-covariances (see Fig. 4b-g) as a fraction of the variance for $P$ ($\sigma_P^2$)**

**and the aridity index ($\overline{E_o}/\overline{P}$) for grid-cells over different latitude ranges (i.e., 90N-60N, 60N-30N, 30N-0 and 0-90S).**

**The colours relate to the water storage capacity $S_{max}$. Note that we have multiplied the covariances by two (see Eq. 2).**

**The vertical grey dashed lines represent thresholds used to separate extremely dry ($\overline{E_o}/\overline{P} \geq 6.0$) and wet ($\overline{E_o}/\overline{P} \leq 0.5$)**

**environments. Note the use of a logarithmic x-axis and scale bar for $S_{max}$.**



625

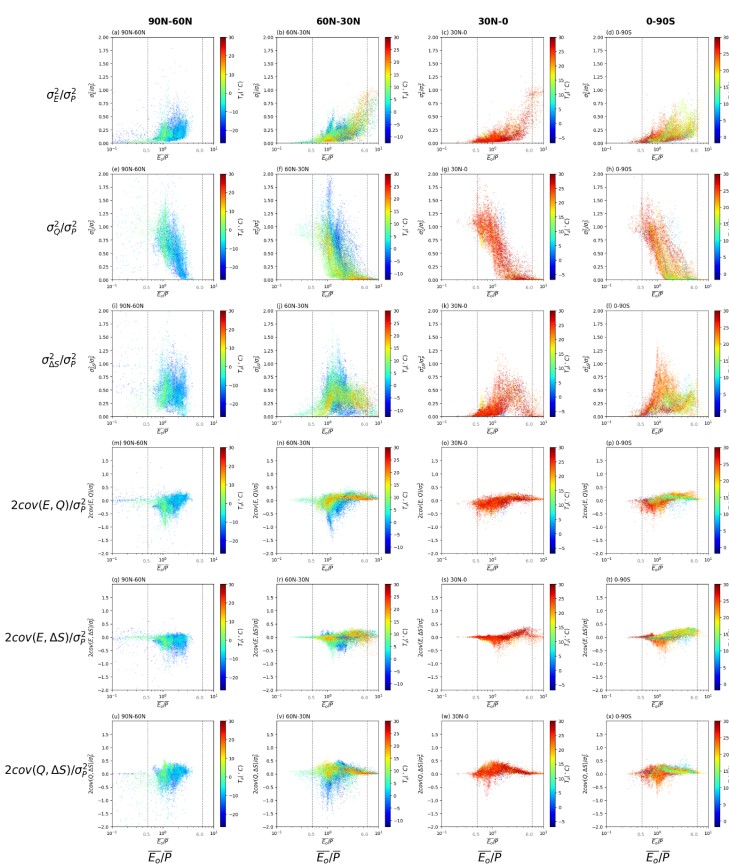

626

**Figure 10. Relation between water cycle variances-covariances (see Fig. 4b-g) as a fraction of the variance for $P$ ($\sigma_P^2$)**

**and the aridity index ($\overline{E_o}/\overline{P}$) for grid-cells over different latitude ranges (i.e., 90N-60N, 60N-30N, 30N-0 and 0-90S).**

**The colours relate to the mean air temperature ($\overline{T_a}$). Note that we have multiplied the covariances by two (see Eq. 2).**

**The vertical grey dashed lines represent thresholds used to separate extremely dry ($\overline{E_o}/\overline{P} \geq 6.0$) and wet ($\overline{E_o}/\overline{P} \leq$**

**0.5) environments.**







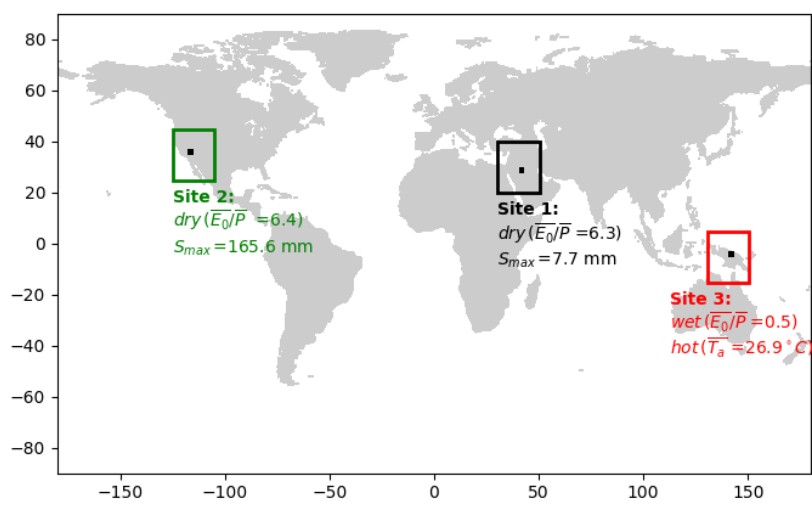


**Figure 11. Locations of three representative grid-cells used as case study sites.**




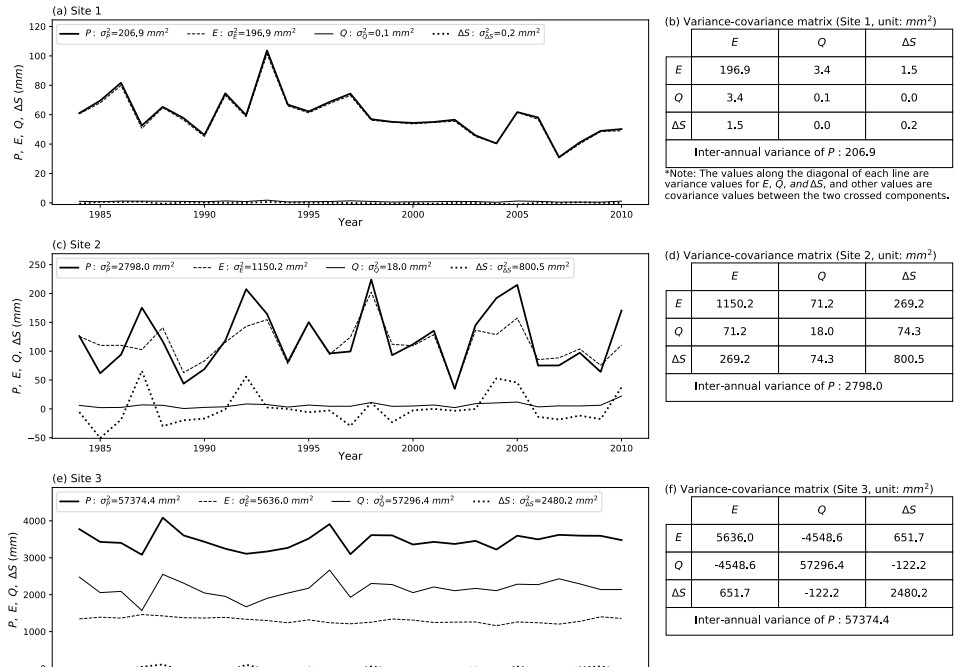


**Figure 12. Inter-annual time series ($P$, $E$, $Q$ and $\Delta S$) and the associated variance-covariance matrix ($E$, $Q$ and $\Delta S$) for case study Sites 1-3. Left column shows time series for (a) Site 1, (c) Site 2 and (e) Site 3, with right column i.e., (b), (d) and (f), the associated variance-covariance matrix for three sites. Note that the covariance values in the tables should be multiplied by two to agree with the variance-covariance balance in Eq. (2).**







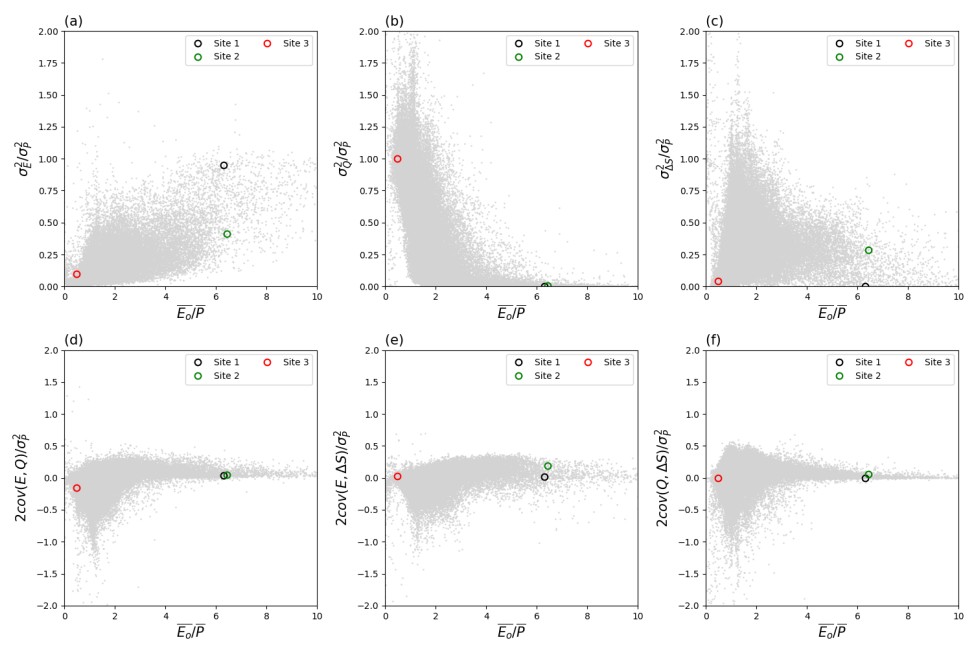


**Figure 13. Location of three case study sites in the water cycle variability space. The grey background dots are from**
**Fig. 8.**






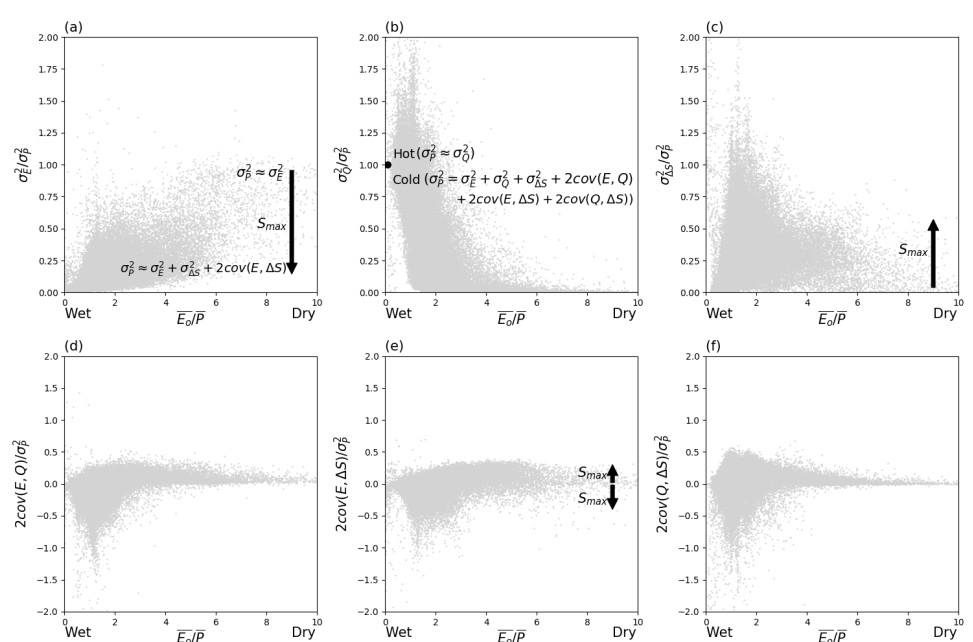


**Figure 14. Synthesis of factors controlling variance partitioning. The arrows denote trends with increasing $S_{max}$. The**
**grey background dots are from Fig. 8.**