# Peer review of "Inter-annual variability of the global terrestrial water cycle"

_Hydrology and Earth System Sciences, 2019_

## Short Comment (SC1) · 24 May 2019

Only 45% of households in Indonesia can access adequate drinking water. This figure is still half of the 2015 Millennium Development Goals (MDGs), where the Indonesian population (86.9%) must be able to access clean and healthy drinking water. This condition contributes to one hundred thousand Indonesian children who die from diarrhea every year. "This is made worse by the presence of global warming which threatens to decrease the quality and quantity of clean water," News : http://www.unair.ac.id/global-warming-ancam-ketersediaan-berita_1314.html

---

## Referee Comment (RC1) · Anonymous Referee #1 · 7 Jun 2019

This is an excellent paper with major implications to our understanding of long term water balance and their climatic and landscape controls.

This kind of work could not have been even just a few years ago, but as more and more reanalysis data become available the ability to do this kind of work and learn from it improves (given the caveat that this is ultimately model generated data, but the best we have).

I have no problems with the analyses that have been done, and the presentation. The authors use monthly data but the analysis is about inter-annual variability, although they do use the monthly data to estimate the storage capacity. I would like to see a categorical statement about this, I found it confusing. This means they only have 28 years of data (28 numbers) - they need to make an assessment/statement about

the implications of this for their estimates of the various statistics, given potential non-stationarities etc.

The main issue that I have with the paper is that (as the authors themselves admit) is the preliminary nature of the discussion and conclusions. The results, to say the least, are quite interesting and intriguing. Without further analysis, one can only speculate. The dependence on storage capacity and temperature are potential clues. This is a concern for me - one solution is to delay the paper until further analysis is done to elucidate these results. It seems the main route to explanations is to use the monthly data that they already have, to see if there is an extension of the variances and especially cross-covariances into the seasonal regime. In other words, I am speculating if the causes of the inter-annual variability lie in the intra-annual variability of the fluxes and the storage, and in the role of vegetation (and soils) buffering the variability in the climate.

For now there is a decision to be made - I am comfortable with going ahead with publication of the current paper (in spite of its preliminary nature) in view of the fact publication of the paper may trigger follow-on research by other research groups as well.

---

## Author Comment (AC1) · 14 Jun 2019

**In the following we use R1C1 (etc) to refer to comment 1 (C1) by referee 1 (R1).**

**Anonymous Referee #1**

R1C1: This is an excellent paper with major implications to our understanding of long term water balance and their climatic and landscape controls.

Response: We thank the anonymous reviewer for the evaluation and positive comment on the manuscript.

R1C2: This kind of work could not have been even just a few years ago, but as more and more reanalysis data become available the ability to do this kind of work and learn from it improves (given the caveat that this is ultimately model generated data, but the best we have).

I have no problems with the analyses that have been done, and the presentation. The authors use monthly data but the analysis is about inter-annual variability, although they do use the monthly data to estimate the storage capacity. I would like to see a categorical statement about this, I found it confusing. This means they only have 28 years of data (28 numbers) - they need to make an assessment/statement about the implications of this for their estimates of the various statistics, given potential non-stationarities etc.

Response: In this initial investigation, we use the CDR (monthly database) and as the reviewer has noted this is an entirely new field of research since global hydrologic reanalysis data has not previously been available. We chose to focus on the inter-annual variability to establish links directly with important earlier work on this topic (e.g., Koster & Suarez, 1999). We plan to extend this work to a seasonal time scale in future research. To eliminate the potential confusion, we will make a categorical statement/review in the revised version of manuscript as the reviewer suggested. Also, another statement about the potential implications of using limited time periods, e.g., non-stationarities, will be added in the revision. Thanks.

R1C3: The main issue that I have with the paper is that (as the authors themselves admit) is the preliminary nature of the discussion and conclusions. The results, to say the least, are quite interesting and intriguing. Without further analysis, one can only speculate. The dependence on storage capacity and temperature are potential clues. This is a concern for me - one solution is to delay the paper until further analysis is done to elucidate these results. It seems the main route to explanations is to use the monthly data that they already have, to see if there is an extension of the variances and especially cross-covariances into the seasonal regime. In other words, I am speculating if the causes of the inter-annual variability lie in the intra-annual variability of the fluxes and the storage, and in the role of vegetation (and soils) buffering the variability in the climate.

Response: We agree with the reviewer that the inter-annual variability will be related to the seasonal (i.e. intra-annual) variations of the fluxes, e.g., the vegetation/soil buffering. As we have noted above (R1C2) we plan to do this in a future work. However, given the initial stage for this type of research we were personally surprised by the complexity of the spatial patterns in the inter-annual variability and sought to publish those before proceeding to a more complex (e.g., seasonal) analysis.

R1C4: For now there is a decision to be made - I am comfortable with going ahead with publication of the current paper (in spite of its preliminary nature) in view of the fact publication of the paper may trigger follow-on research by other research groups as well.

Response: We appreciate the comments of the reviewer.

---

## Referee Comment (RC2) · Rene Orth (Referee) · 19 Jun 2019

Review of Dongqin and Roderick "Inter-annual variability of the global terrestrial cycle"

This study investigates the propagation of precipitation variability into the water cycle, i.e. into variations of runoff, evapotranspiration, and of storage changes. The authors show that this is mostly controlled by temperature (in wet regions), long-term aridity (in transitional regions), and by soil water storage capacity (in dry regions). Further, the results illustrate that the corresponding partitioning is different from the partitioning of mean precipitation into the means of these water cycle variables.

————————-

Recommendation: I think the paper requires major revisions.

The analysis is very interesting and provides new and fundamental insights into large-scale land surface hydrology. Related variability analyses are still not commonly done due to a lack of reliable data and underlying theory. This study can foster theory development in this area, and it underlines the importance of continuous improvement of the just-emerging global hydrological re-analysis datasets. Therefore I would be happy to see it published in HESS, but after some general revisions:

(1) Next to the consideration of the soil water storage capacity and the mean temperature to explain variations in the partitioning of precipitation variability, I am missing the inclusion of vegetation type as an explanatory variable. It might have strong implications on evapotranspiration variability, and therefore also on runoff and storage variabilities.

(2) I agree with the authors that comprehensive hydrological reanalysis datasets are lacking, and the CDR dataset is an important contribution in that respect. Further, I appreciate the effort they make to validate the applicability of the dataset in the context of this study. However, also the CDR dataset is (necessarily) based on a model and hence it is not clear that the reported relationships are operating in nature, and not only in this model. To address this issue, I would like to see the key analyses from this study repeated with the state-of-the-art ERA5 reanalysis, which should be superior to ERA-Interim also in terms of land surface representation.

(3) I appreciate the idea of investigating the influence of the soil water storage capacity and the mean temperature on the variability partitioning. However, I think parts of the conclusions drawn by the authors from Figures 8-10 are not supported by the data. For example, I cannot see in Figure 10 that the temperature influence is particularly strong in very wet regions. Rather, to me it seems to be strong in moderately wet and dry regions (Fig 10b,d,f,h,j,l,n,p). Further, also the aridity limit of 6 which the authors suggest in their interpretation of the results in Figure 9, is arbitrary and not supported by the actual results. Storage capacity is obviously having an influence already for aridity values above 2-3 (Fig. 9b,c,f,j,k). Overall, in these Figures there are many interesting patterns but the authors focus only on few sub-plots and limit their interpretation to these. Therefore, I suggest to either show less information/sub-plots there, or to develop explanations also for patterns emerging within other sub-plots.

(4) The paper contains (too) many figures, which is diluting the main message(s), I feel. For example, Figures 1 and 2 could be merged, Figure 5 could be moved to the supplementary material, Figure 13 could be merged into Figure 8. The authors might have further ideas to reduce the amount of figures. Moreover, I do not really understand the difference between Figures 7 and 8, and why both are needed.

I do not wish to remain anonymous - René Orth.
* * *
Specific comments:

line 8: Equation 2 not introduced yet

line 13: It should be 'variabilities'.

line 15: Some word is missing towards the end of the line

lines 35-39: Orth & Destouni (2018) might be relevant in this context and could be cited.

line 37: Not sure I get the point here.

lines 106-118: Please clarify that what you are determining here is actually not the soil water storage capacity, but rather the active range within which the soil moisture varies.

lines 157-163: I would recommend to replace the LandFluxEVAL and the Jung et al. datasets with more recent gridded ET datasets such as the Jung et al. 2019 dataset and the GLEAM dataset (Martens et al. 2017).

line 180: Gudmundsson et al. (2016) might be relevant in this context and could be cited.

line 181: What is meant by seasonality here? I thought you are considering annual data? In general, I think the considered temporal and spatial scales and resolution need to be more clearly stated and motivated at the beginning of the manuscript. Also, the role of these decisions on the results could be discussed.

line 252/253: I could not find this discussion in section 5!? Would be important to explain these discrepancies, though.

line 327 & 333: 'leaving very limited variance' - not really true given your statement in lines 385-387

lines 403-405: I cannot see this from Figure 8.

Section 5: Overall a bit lengthy with too much summarizing, I think. Could be shorter, and more concise.

Figure 3: Why are there data points outside the physically plausible range?

Figure 4: Many values seem to be cut at 10 as this is the end of the color bar. You could use log scale here for the color bar.

References:

Gudmundsson, L., P. Greve, and S. I. Seneviratne, 2016: The sensitivity of water availability to changes in the aridity index and other factors—A probabilistic analysis in the Budyko space, Geophys. Res. Lett. 43 (13), 6985-6994.

Jung, M., S. Koirala, U. Weber, K. Ichii, F. Gans, G. Camps-Valls, D. Papale, C. Schwalm, G. Tramontana, and M. Reichstein, 2019: The FLUXCOM ensemble of global land-atmosphere energy fluxes. Scientific Data, 6 (74).

Martens, B., D. G. Miralles, H. Lievens, R. van der Schalie, R. A. M. de Jeu, D. Fernández- Prieto, H. E. Beck, W. A. Dorigo, and N. E. C. Verhoest, 2017: GLEAM v3: satellite-based land evaporation and root-zone soil moisture, Geosci. Model Dev. 10, 1903–1925.

Orth, R., and G. Destouni, 2018: Drought reduces blue-water fluxes more strongly than green-water fluxes in Europe. Nature Communications, 9, 3602, doi: 10.1038/s41467-018-06013-7

---

## Referee Comment (RC3) · Anonymous Referee #3 · 24 Jun 2019

This study tries to partition the inter-annual variability in precipitation (P), i.e., the source term in terrestrial water cycle, into variabilities in three sink terms in terrestrial water cycle (ET, Q, $\Delta$S), and then to relate the partitioning of variabilities to various factors like temperature, aridity, and storage capacity. I think this type of study at global scale is rather new, if not first of its kind at global scale, and thus very interesting to the hydrology community. This is the case mostly because there has been a lack of "hydrologic reanalysis" (CDR) for such kind of analysis in the first place. At the same time, this effort couldn't fully answer many of the questions set forth at the beginning, leaving perhaps "more questions than answers" (as phrased by another referee). The authors have done a solid amount of thorough analysis and experiments toward the questions of interest and these analyses are also well designed too.

[Figure]

Overall I consider this manuscript of good quality, both scientifically and technically, and thus publishable in HESS with several concerns addressed:

My primary concern is there is a lack of general "signal-to-noise" discussions to better inform readers to what extent the findings are significant signals from the underlying data (CDR, Zhang et al., 2018) and how much of it could be due to data uncertainties (or possible artifacts due to how the data is produced). For example, the ET products that went into the CDR (satellite products, reanalysis, etc.) share some similarity in their production methods (e.g., Penman-Monteith or Priestley-Taylor type of schemes). Such similarity may limit the variability of ET in CDR. Of course, the plants do apply a strong filter on the inter-annual variability based on their survival need. Such uncertainty analysis may be difficult but I think some qualitative and general assessment would be very beneficial.

Also, at the scale of the CDR (0.5 degree), I would say the partitioning is more complicated than just a result of several factors. The horizontal transport of water, seasonality, local water use, etc., can add a lot of noise. I wouldn't say it is not possible to do it at 0.5 degree, but it would probably be less noisy at a slightly coarser scale. Also, there could be much more controlling factors for the partitioning than being investigated, e.g., land cover/land use, LAI, topography.

Finally, given that this study does tend to raise more questions than answers, I feel the authors should provide some more insights on what we can do from the analysis and findings in this study. What can we do with the numbers concluded here? Validating models? Improving single models like Budyko? Hydrologic/water risk analysis? Climate system behavior/sensitivity and hydrologic impacts of climate changes? And how can we improve our understanding in the future? What kind of new data at what scales would be critical to answering such questions? I feel this paper is incomplete without offering some of such insights.
* * *
[Figure]

230, 2019.

---

## Author Comment (AC2) · 16 Jul 2019

**In the following we use R2C1 (etc) to refer to comment 1 (C1) by referee 2 (R2).**

**Dr René Orth**

R2C1: Review of Dongqin and Roderick "Inter-annual variability of the global terrestrial cycle" This study investigates the propagation of precipitation variability into the water cycle, i.e. into variations of runoff, evapotranspiration, and of storage changes. The authors show that this is mostly controlled by temperature (in wet regions), long-term aridity (in transitional regions), and by soil water storage capacity (in dry regions). Further, the results illustrate that the corresponding partitioning is different from the partitioning of mean precipitation into the means of these water cycle variables.

————————-

Recommendation: I think the paper requires major revisions.

The analysis is very interesting and provides new and fundamental insights into large- scale land surface hydrology. Related variability analyses are still not commonly done due to a lack of reliable data and underlying theory. This study can foster theory development in this area, and it underlines the importance of continuous improvement of the just-emerging global hydrological re-analysis datasets. Therefore I would be happy to see it published in HESS, but after some general revisions.

Response: We thank Dr René Orth for the evaluation and comments on the manuscript.

R2C2: (1)    Next to the consideration of the soil water storage capacity and the mean temperature to explain variations in the partitioning of precipitation variability, I am missing the inclusion of vegetation type as an explanatory variable. It might have strong implications on evapotranspiration variability, and therefore also on runoff and storage variabilities.

Response: We agree with Dr René Orth that the inter-annual variability might be related to the other factors, e.g., vegetation type. However, given the fact that this is a new approach and the research is exploratory, we focused on relating the inter-annual variability with the most general hydrologic factors (i.e., the air temperature as a surrogate for snow/ice and water storage capacity). We expect to extend the current work to a more complete analysis (e.g., relation to vegetation) in future research.

R2C3: (2)    I agree with the authors that comprehensive hydrological reanalysis datasets are lacking, and the CDR dataset is an important contribution in that respect. Further, I appreciate the effort they make to validate the applicability of the dataset in the context of this study. However, also the CDR dataset is (necessarily) based on a model and hence it is not clear that the reported relationships are operating in nature, and not only in this model. To address this

issue, I would like to see the key analyses from this study repeated with the state-of-the-art ERA5 reanalysis, which should be superior to ERA-Interim also in terms of land surface representation.

Response: One of the reasons for developing hydrologic reanalyses like the CDR (and hopefully other forthcoming hydrologic reanalyses) is that the broader hydrologic community were not satisfied with atmospheric reanalyses (e.g., ERA/NCEP). In that context, CDR is "hydrologic-centric" as opposed to ERA which is "atmospheric-centric". In creating the original CDR, they actually used evaporation from the ERA-interim (as one of 8 different $E$ products, see Table 1 in Zhang et al. 2018, HESS). To replace $E$ from ERA-interim with that from ERA5 would require us to completely re-do the CDR data assimilation but that is well beyond the scope of this work.

In terms of relating the CDR to the real world there is ample evidence that it is suitable for the analysis conducted here including:

(i)    The enforcement of basic hydrologic concepts (mass balance).

(ii)   The numerous tests of CDR reported in the original Zhang et al 2018 HESS publication (that are summarized on lines 134-139 of the HESSD manuscript). Those tests include a (successful) comparison of CDR runoff to observations of monthly runoff at 165 medium size basins and 862 small basins. In fact, the assessment of CDR in the original paper was quite comprehensive as you would expect.

(iii)  We have augmented those extensive original tests by independently comparing monthly $E$ with FLUXNET tower data at 32 sites which confirmed that the CDR captured the general seasonal cycle in both $P$ and $E$ at those 32 sites (Fig. S2, S3, S4, Table S1). We also used the same FLUXNET data to compare the variability in $P$ with variability in $E$ (Fig. S5).

(iv)   We further compared CDR $E$ with two gridded $E$ databases that are not included in the source databases of CDR (LandFluxEval, MPI, see lines 157-163 in the manuscript and Fig. S6, S7) and the comparison was satisfactory.

(v)    We compared how the standard deviation for $E$ and the mean for $E$ are related in the CDR (Fig. 6) and compared that with the same relations in LandFluxEval and MPI (Fig. S8). Those two comparisons were satisfactory.

(vi)   The mean water cycle ($P$, $E$, $Q$) in CDR was shown to be consistent with the long-standing Budyko framework (Fig. 3).

(vii)  The CDR data were consistent with the Koster & Suarez (1999) theory in the limit of sites that have limited water storage (Fig. 7).

That is a very comprehensive assessment.

In summary, there will always be new databases for comparison but the above tests led us to conclude that the CDR was at least suitable for an exploratory analysis of water cycle variances using the completely new "variance balance approach" (Eqn 2) described in the manuscript.

R2C4: (3) I appreciate the idea of investigating the influence of the soil water storage capacity and the mean temperature on the variability partitioning. However, I think parts of the conclusions drawn by the authors from Figures 8-10 are not supported by the data. For example, I cannot see in Figure 10 that the temperature influence is particularly strong in very wet regions. Rather, to me it seems to be strong in moderately wet and dry regions (Fig 10b,d,f,h,j,l,n,p). Further, also the aridity limit of 6 which the authors suggest in their interpretation of the results in Figure 9, is arbitrary and not supported by the actual results. Storage capacity is obviously having an influence already for aridity values above 2-3 (Fig. 9b,c,f,j,k). Overall, in these Figures there are many interesting patterns but the authors focus only on few sub-plots and limit their interpretation to these. Therefore, I suggest to either show less information/sub-plots there, or to develop explanations also for patterns emerging within other sub-plots.

Response: We accept that Fig. 10 is hard to interpret. On reading the reviewers comments and going over the manuscript we realize the problem was that we did not explicitly indicate the relevant panels (i.e., a, b, c, ….). That was on oversight correctly identified by the reviewer. In general, the data in Fig. 10 was not particularly revealing (i.e., a negative result) but we actually focused the discussion to the first two columns but we did not identify them properly. In response, we propose to replace the original text with the following:

*"To understand the potential role of snow/ice in modifying the variance partitioning, we repeat the previous analysis (Fig. 9) but here we use the mean annual air temperature ($\overline{T_a}$) to colour the grid-cells to crudely identify the presence of snow/ice (Fig. 10). Most of the variations at polar latitudes in the northern hemisphere (panels in the first and second columns of Fig. 10) is associated with low air temperature (e.g., $\overline{T_a} < 0$ ˚C), making the results associated with high air temperature (e.g., $\overline{T_a} > 10$ ˚C in third and fourth columns of Fig. 10) show less scatters. That pattern is particularly obvious in extremely wet environment ($\overline{E_o}/\overline{P} \leq 0.5$), where the ratio $\sigma_Q^2/\sigma_P^2$ is close to 1.0 when $\overline{T_a}$ is high (e.g., $\overline{T_a} > 10$ ˚C, Fig. 10gh) but shows lots of scatters when $\overline{T_a}$ is low (e.g., $\overline{T_a} < 0$ ˚C, Fig. 10ef). This indicates that in extremely wet environment, when $\overline{T_a}$ is high, $\sigma_P^2$ is almost completely partitioned into $\sigma_Q^2$ (e.g., $\overline{E_o}/\overline{P} \leq 0.5$ and $\overline{T_a} > 10$ ˚C in the third and fourth columns of Fig. 10). However, when $\overline{T_a}$ is low in extremely wet environment, there are substantial variations in all variance-covariance components (e.g., $\overline{E_o}/\overline{P} \leq 0.5$ and $\overline{T_a} < 0$ ˚C in the first and second columns of Fig. 10). That result indicates the complexity of variance partitioning associated with the presence of snow/ice."*

R2C5: (4) The paper contains (too) many figures, which is diluting the main message(s), I feel. For example, Figures 1 and 2 could be merged, Figure 5 could be moved to the supplementary material, Figure 13 could be merged into Figure 8. The authors might have further ideas to reduce the amount of figures. Moreover, I do not really understand the difference between Figures 7 and 8, and why both are needed.

I do not wish to remain anonymous - René Orth.

Response: We respect the reviewer's opinion that we have too many figures – this is always a hard balance to get right to everyone's satisfaction. We can easily combine Fig. 1 and 2 as suggested, or alternatively, we could move Fig. 1 to the supporting material. We can also move Fig. 5 to the supporting material as suggested. However, we do not think Fig. 13 should be merged into Fig. 8 since the two figures belong to different sections (Fig. 8 for the relation between variance partitioning and aridity section, Fig. 13 for the case study section). Fig. 7 is a direct link to previous work while Fig. 8 is the variance partitioning in the CDR database. Hence while these two figures are similar, they make separate independent contributions to the manuscript.
* * *
Specific comments:

R2C6: line 8: Equation 2 not introduced yet line 13: It should be 'variabilities'.

Response: We will modify these texts in the revised version of manuscript. Thanks.

R2C7: line 15: Some word is missing towards the end of the line

Response: We have checked line 15 and did not find missing words?

R2C8: lines 35-39: Orth & Destouni (2018) might be relevant in this context and could be cited.

Response: The reference will be cited in the revised manuscript.

R2C9: line 37: Not sure I get the point here.

Response: We mean that droughts and floods are typical extremes but that hydrologic variability encompasses more than just droughts and floods, i.e., hydrologic variability occurs across all time-space scales.

R2C10: lines 106-118: Please clarify that what you are determining here is actually not the soil water storage capacity, but rather the active range within which the soil moisture varies.

Response: Yes, exactly. We can modify the text to make this explicit.

R2C11: lines 157-163: I would recommend to replace the LandFluxEVAL and the Jung et al. datasets with more recent gridded ET datasets such as the Jung et al. 2019 dataset and the GLEAM dataset (Martens et al. 2017).

Response: The reason we chose the LandFluxEVAL and MPI databases is that they are among the most widely used and validated $E$ data that were also not used to develop the CDR database. We do not think adding a comparison to the latest GLEAM database would be as useful since an earlier version of GLEAM (v2a) was actually an input to the data assimilation scheme used to construct the CDR (see Table 1 in Zhang et al., 2018, HESS). Instead it would be better to re-do the CDR data assimilation incorporating the latest GLEAM database but that is well beyond the scope of this work. (Also see R2C3 for similar comments about ERA.) We could replace the MPI we used with the updated database (Jung et al., 2019) but we do not see how that would alter the results.

R2C12: line 180: Gudmundsson et al. (2016) might be relevant in this context and could be cited.

Response: The reference will be cited in the revised manuscript. Thanks.

R2C13: line 181: What is meant by seasonality here? I thought you are considering annual data? In general, I think the considered temporal and spatial scales and resolution need to be more clearly stated and motivated at the beginning of the manuscript. Also, the role of these decisions on the results could be discussed.

Response: Yes, we are using annual data. But we know that differences of the intra-year seasonal timing (phase) of precipitation and $E_o$ do have an effect on the annual water balance (as per the seminal work by Chris Milly in the early 1990s.). We will make this more clear in the revised version.

R2C14: line 252/253: I could not find this discussion in section 5!? Would be important to explain these discrepancies, though.

Response: Thanks for pointing this oversight out. The underlying scientific issue here is that the original Koster and Suarez (1999) work assumed no change in long-term storage. In that sense the original results of Koster and Suarez (1999) can be seen as an upper limit and any variance in storage can only reduce the partitioning of variability in $P$ to variability in $E$ under dry conditions (Fig. 7). We will add a short discussion on this in the revised manuscript.

R2C15: line 327 & 333: 'leaving very limited variance' - not really true given your statement in lines 385-387

Response:

324 We show the $P$, $E$, $Q$ and $\Delta S$ time series along with the relevant variances and covariances in Fig. 12. Starting

325 with the two dry sites, at the site with low storage capacity (Site 1), the time series shows that $E$ closely follows

326 $P$ leaving annual $Q$ and $\Delta S$ close to zero (Fig. 12a). The variance of $P$ ($\sigma_P^2 = 206.9$ mm$^2$) is small and almost

327 completely partitioned into the variance of $E$ ($\sigma_E^2 = 196.9$ mm$^2$), leaving very limited variance for $Q$, $\Delta S$ and all

328 three covariance components (Fig. 12b). At the site with high storage capacity (Site 2), $E$, $Q$ and $\Delta S$ do not simply

329 follow $P$ (Fig. 12c). As a consequence, the variance of $P$ ($\sigma_P^2 = 2798.0$ mm$^2$) is shared between $E$ ($\sigma_E^2 = 1150.2$

330 mm$^2$), $\Delta S$ ($\sigma_{\Delta S}^2 = 800.5$ mm$^2$) and their covariance component ($2cov(E, \Delta S) = 538.4$ mm$^2$, Fig. 12d). Switching

331 now to the remaining wet and hot site (Site 3), $Q$ closely follows $P$, with $\Delta S$ close to zero and $E$ showing little

332 inter-annual variation (Fig. 12e). The variance of $P$ ($\sigma_P^2 = 57374.4$ mm$^2$) is relatively large and almost completely

333 partitioned into the variance of $Q$ ($\sigma_Q^2 = 57296.4$ mm$^2$), leaving very limited variance for $E$ and $\Delta S$ and the three

334 covariance components (Fig. 12f). We also examined numerous other sites with similar extreme conditions as the

335 three case study sites and found the same basic patterns as reported above.

The text here refers to the site-based case studies (line 327 – Fig. 12a – Site 1; line 333 – Fig. 12 f – Site 3) while the later text (lines 385-387) refers to the general pattern across all grid-boxes, i.e., Fig. 4. Perhaps we can correct this misunderstanding by rewriting lines 385-387 to indicate the relevant figures as follows:

*"Hence we were initially surprised that the inter-annual variability of water storage change ($\sigma_{\Delta S}^2$) is typically larger than the inter-annual variability of evapotranspiration ($\sigma_E^2$) (cf. Fig. 4b and 4d). Moreover, the covariance components are also prominent and can be negative (Fig. 4efg), which means that it is possible for the variability in the sinks (e.g., $\sigma_Q^2$, $\sigma_{\Delta S}^2$) can actually exceed the variability in the source ($\sigma_P^2$) (Eq. 2)."*

R2C16: lines 403-405: I cannot see this from Figure 8.

Response: Agreed. That was our mistake. The reference to Fig. 8 should be to Fig. 4 (global pattern of water cycle variability) and we will revise that in the revision.

R2C17: Section 5: Overall a bit lengthy with too much summarizing, I think. Could be shorter, and more concise.

Response: We will read and revise the Section 5 carefully and make it more concise accordingly in the revision as per the comments of both R2 and R3.

R2C18: Figure 3: Why are there data points outside the physically plausible range?

Response: We assume you mean points with $E$ exceeding $P$? This is possible in for example, regions with run-on, or irrigation. We have further investigated those points and also find that some of them come from the parts of Greenland that had not been masked out (Fig. 1). Non-steady state conditions, e.g. long-term changes in storage can also lead to $E$ exceeding $P$.

R2C19: Figure 4: Many values seem to be cut at 10 as this is the end of the color bar. You could use log scale here for the color bar.

Response: Yes, the scale for $P$ in Fig. 4a is saturated with the maximum value of the color bar 10000. The reason we chose 10000 as the limit was to show the patterns for both the relative high (e.g., $\sigma_P^2$, $\sigma_Q^2$ and $\sigma_{\Delta S}^2$) and low variabilities (e.g., $\sigma_E^2$, 2cov($E$, $\Delta S$)). We can modify by using a log scale to address this comment.

R2C20: References:

Gudmundsson, L., P. Greve, and S. I. Seneviratne, 2016: The sensitivity of water avail- ability to changes in the aridity index and other factorsâ˘A˘A probabilistic analysis in the Budyko space, Geophys. Res. Lett. 43 (13), 6985-6994.

Jung, M., S. Koirala, U. Weber, K. Ichii, F. Gans, G. Camps-Valls, D. Papale, C. Schwalm, G. Tramontana, and M. Reichstein, 2019: The FLUXCOM ensemble of global land-atmosphere energy fluxes. Scientific Data, 6 (74).

Martens, B., D. G. Miralles, H. Lievens, R. van der Schalie, R. A. M. de Jeu, D. Fernández-Prieto, H. E. Beck, W. A. Dorigo, and N. E. C. Verhoest, 2017: GLEAM v3: satellite-based land evaporation and root-zone soil moisture, Geosci. Model Dev. 10, 1903–1925.

Orth, R., and G. Destouni, 2018: Drought reduces blue-water fluxes more strongly than green-water fluxes in Europe. Nature Communications, 9, 3602, doi: 10.1038/s41467- 018-06013-7

Response: We appreciate Dr René Orth for listing all the reference mentioned above in the comments, and we will read and cite these reference accordingly in the revised manuscript. Thanks.

---

## Author Comment (AC3) · 17 Jul 2019

**In the following we use R3C1 (etc) to refer to comment 1 (C1) by referee 3 (R3).**

**Anonymous Referee #3**

R3C1: This study tries to partition the inter-annual variability in precipitation (P), i.e., the source term in terrestrial water cycle, into variabilities in three sink terms in terrestrial water cycle (ET, Q, ΔS), and then to relate the partitioning of variabilities to various factors like temperature, aridity, and storage capacity. I think this type of study at global scale is rather new, if not first of its kind at global scale, and thus very interesting to the hydrology community. This is the case mostly because there has been a lack of "hydrologic reanalysis" (CDR) for such kind of analysis in the first place. At the same time, this effort couldn't fully answer many of the questions set forth at the beginning, leaving perhaps "more questions than answers" (as phrased by another referee). The authors have done a solid amount of thorough analysis and experiments toward the questions of interest and these analyses are also well designed too.

Overall I consider this manuscript of good quality, both scientifically and technically, and thus publishable in HESS with several concerns addressed.

Response: We agree that this is a first-of-its-kind study and thank the referee for the encouraging positive comments on the manuscript.

R3C2: My primary concern is there is a lack of general "signal-to-noise" discussions to better inform readers to what extent the findings are significant signals from the underlying data (CDR, Zhang et al., 2018) and how much of it could be due to data uncertainties (or possible artifacts due to how the data is produced). For example, the ET products that went into the CDR (satellite products, reanalysis, etc.) share some similarity in their production methods (e.g., Penman-Monteith or Priestley-Taylor type of schemes). Such similarity may limit the variability of ET in CDR. Of course, the plants do apply a strong filter on the inter-annual variability based on their survival need. Such uncertainty analysis may be difficult but I think some qualitative and general assessment would be very beneficial.

Response: The CDR uses a formal data assimilation scheme based on mass balance that weights the various inputs, and thereby produces uncertainty estimates for each variable ($P$, $E$, $Q$, $\Delta S$). The original paper (Zhang et al., 2018 HESS) includes a formal assessment of the sensitivity of $P$, $E$, $Q$ over large regions (continents, basins) using the coefficient of variation (see original Figures 2, 3, 4, 5, 6, 7 in Zheng et al., 2018 HESS). We actually followed from that work and used those uncertainty estimates (lines 122-130) to identify and mask out regions where we judged the uncertainty to be large relative to the magnitude of the fluxes. This screening procedure removed most grid-boxes from the Himalayas, Sahara Desert and Greenland (see Fig. 1).

Secondly, while it is true that some of the products might share similarity in producing, for example, $E$ (Penman-Monteith, Priestley-Taylor as the examples noted by the reviewer) the data assimilation is a comprehensive approach that includes all available estimates of $P$, $E$, $Q$ and $\Delta S$ at each grid box. With mass balance enforced, the CDR estimates represent a composite product that is designed to avoid bias of the type described by the reviewer as much as possible by using all available estimates of the hydrologic fluxes. As we have described in a response to Reviewer 2 (see R2C3), the CDR has been extensively validated in the original publication. In that context, our goal was not to assess the CDR, but rather to use it for this "first-of-a-kind" study on the sources and sinks of inter-annual hydrologic variability.

In summary, with the many individual validations of the CDR in the original paper (Zhang et al., 2018 HESS) augmented by those in our manuscript, our results are based on the best available hydrologic reanalyses. In terms of the remaining uncertainty from the CDR data, this is beyond the scope of the current study. Despite that, the general approach in our manuscript will remain and the results will be fine-tuned over the coming years as the hydrologic community develops and uses their own reanalyses. We will add words to that effect in the revised version of the manuscript.

R3C3: Also, at the scale of the CDR (0.5 degree), I would say the partitioning is more complicated than just a result of several factors. The horizontal transport of water, seasonality, local water use, etc., can add a lot of noise. I wouldn't say it is not possible to do it at 0.5 degree, but it would probably be less noisy at a slightly coarser scale. Also, there could be much more controlling factors for the partitioning than being investigated, e.g., land cover/land use, LAI, topography.

Response: We agree with the reviewer that the partitioning is complex and could be related to the other factors, e.g., land cover/land use, LAI and horizontal transport of water due to topography, etc. In this first-of-a-kind analysis we chose to focus on the zero'th order physical factors (storage capacity, snow/ice) at the CDR data resolution (0.5 degree), but we fully expect more detailed analysis to follow, e.g., vegetation plant-based variables as discussed by the reviewer.

R3C4: Finally, given that this study does tend to raise more questions than answers, I feel the authors should provide some more insights on what we can do from the analysis and findings in this study. What can we do with the numbers concluded here? Validating models? Improving single models like Budyko? Hydrologic/water risk analysis? Climate system behavior/sensitivity and hydrologic impacts of climate changes? And how can we improve our understanding in the future? What kind of new data at what scales would be critical to answering such questions? I feel this paper is incomplete without offering some of such insights.

Response: We thank the reviewer for the constructive suggestion on the insights of this study. This is awkward – what the reviewer is asking for is an extended discussion while reviewer 2 has asked for less discussion. In the revision we will try our best to find a balance and set out what can be learnt. To respond in more detail, what we have learnt from undertaking the study

is that; (i) partitioning of hydrologic variability does not follow partitioning of the mean, (ii) the long-ignored covariances play a critical role in hydrologic partitioning, especially in biologically productive environments (aridity index ~1), (iii) and because of those covariances there will be no simple translation of changes in the variability of $P$ into changes in the variability of $E$, $Q$, $\Delta S$. We also expect that in future we will be able to extract generic signatures of hydrologic variability (e.g., Fig. 8) that can be used to assess the simulation of variability in models. In response to this important point made by the reviewer we intend to carefully revise the discussion and add some of the potential implications as requested by the reviewer.

---

## Author Response (AR3)

**Response to Editor**

Comments to the Author:

Dear authors, three reviewers have given feedback on your manuscript. The reviewers give generally very positive feedbacks and state they were intrigued by the analysis and results. All find that it is a timely and valuable contribution to the field of global hydrology. The paper is well structured and written. The reviewers have also given constructive feedback and criticism and you have addressed several of those comments in your response.

I agree with the assessment of the reviewers on the merit and novelty of the presented analysis. I would however like to emphasize one point: All of the reviewers comment, in one way or the other, on the fact that the results hinge upon the correctness of the CDR dataset. In your response you emphasize how carefully the CDR dataset was developed and validated, and also your own efforts to validate e.g. the standard deviation of E. I appreciate this. However, some of the standard variations in the dataset are not yet validated. I agree with reviewer #2 (René Orth) that a cross-validation would be desirable to learn whether the observed variance patterns are a property of the CDR dataset or hold with other datasets. At the very least, and since the main message of the manuscript is a call for investigation into the causes of the observed hydroclimatic variability, the discussion should more than now acknowledge to that fact that any efforts towards validation of those patterns are equally warranted.

Please submit the revised manuscript, with changes highlighted, together with a point by point response to all of the reviewers comments.

I thank both the authors and reviewers for the constructive discussion and look forward to the revised manuscript,

Anke Hildebrandt.

Response: We thank the editor for the evaluation and comment on the manuscript. As suggested by the editor and reviewers, we have carefully read and revised the manuscript accordingly as well as conducted a point-by-point response to all the comments.

The main comment here is a further cross-validation of the CDR data results based on atmospheric reanalysis (e.g., the state-of-the-art ERA5 dataset). As suggested by both editor and R2, in this response we report a comparison of the CDR ($P$, $E$, $Q$ and $\Delta S$) with the same from the recently released ERA5. We found $P$ to be similar in both CDR and ERA5, but we found $E$ and $Q$ to be generally **much** higher in ERA5 compared to CDR (please see details in response to R2C3). As a consequence, in ERA5 we found that the sum of $E$ and $Q$ regularly exceeded $P$ by large amounts. For example, in the Amazon, $E$ and $Q$ exceeded $P$ by up to 1000 mm each and every year. So over a 27 year period, the predicted decline in storage in the Amazon region embedded in ERA5 approached 27000 mm (27 m)! This represents a major problem in the mass balance (or a lack of mass balance) in the ERA5 reanalysis and is physically not plausible. In contrast, over ice covered regions (e.g., Greenland), the hydrologic balance implied a continuing gain in storage of roughly similar magnitudes (i.e., 27 m in 27 years). Again, this is also physically not plausible.

Though the ERA5 is the state-of-the-art atmospheric reanalysis, we concluded that there was a major problem with the hydrologic (mass) balance and that the "atmospheric-centric" ERA5 database was not yet suitable for use in hydrologic studies.

As suggested by the editor, we also added the statement about the importance towards further improvement and validation of the patterns obtained in this manuscript in the revised manuscript.

Another important point raised by the reviewers (R2, R3) were (divergent) criticisms of the summary sections of the original manuscript. After carefully looking at comments from R2 and R3 and the structure and content of the original manuscript, we concluded that the underlying problem was that the original Discussion and Conclusions were repetitive and generally not well formulated. In response, we decided to combine the original sections (sections 5 and 6) into a new single section 5 (Discussion and Conclusions), and have streamlined the text accordingly by integrating the comments by reviewers. We believe that this has made the summary section more concise and that this change has substantially improved the manuscript.

We sincerely appreciate both the editor and reviewers for constructive suggestions and comments on the manuscript.

**Response to Referee #1 (Anonymous)**

**In the following we use R1C1 (etc) to refer to comment 1 (C1) by referee 1 (R1).**

R1C1: This is an excellent paper with major implications to our understanding of long term water balance and their climatic and landscape controls.

Response: We thank the anonymous reviewer for the evaluation and positive comment on the manuscript.

R1C2: This kind of work could not have been even just a few years ago, but as more and more reanalysis data become available the ability to do this kind of work and learn from it improves (given the caveat that this is ultimately model generated data, but the best we have).

I have no problems with the analyses that have been done, and the presentation. The authors use monthly data but the analysis is about inter-annual variability, although they do use the monthly data to estimate the storage capacity. I would like to see a categorical statement about this, I found it confusing. This means they only have 28 years of data (28 numbers) - they need to make an assessment/statement about the implications of this for their estimates of the various statistics, given potential non-stationarities etc.

Response: In this initial investigation, we use the CDR (monthly database) and as the reviewer has noted this is an entirely new field of research since global hydrologic reanalysis data has not previously been available. We chose to focus on the inter-annual variability to establish links directly with important earlier work on this topic (e.g., Koster & Suarez, 1999). We plan to extend this work to a seasonal time scale in future research. To eliminate the potential confusion, we made a statement as the reviewer suggested in the revised version of manuscript:

(Lines 100-101): "*In this study we focus on the inter-annual variability and the monthly water*
*cycle variables (P, E, Q and ΔS) are aggregated to annual totals.*". Also, another statement
about the limitations of 27-year study period has be added in the revision (Lines 457-460):
"*The CDR is one of the first dedicated hydrologic reanalysis databases and includes data for*
*a 27-year period. Accordingly, we could only examine hydrologic variability over this*
*relatively short period. Further, we expect future improvements and modifications as the*
*hydrologic community seeks to further develop and refine these new reanalysis databases.*".
Thanks.
R1C3: The main issue that I have with the paper is that (as the authors themselves admit) is the
preliminary nature of the discussion and conclusions. The results, to say the least, are quite
interesting and intriguing. Without further analysis, one can only speculate. The dependence
on storage capacity and temperature are potential clues. This is a concern for me - one solution
is to delay the paper until further analysis is done to elucidate these results. It seems the main
route to explanations is to use the monthly data that they already have, to see if there is an
extension of the variances and especially cross-covariances into the seasonal regime. In other
words, I am speculating if the causes of the inter-annual variability lie in the intra-annual
variability of the fluxes and the storage, and in the role of vegetation (and soils) buffering the
variability in the climate.

Response: We agree with R1 about the likely importance of the seasonal (i.e. intra-annual)
cycle to further explain these results. However, given the new approach developed in this
manuscript we deliberately chose to publish the somewhat simpler inter-annual results first.
Please also see R1C2.
R1C4: For now there is a decision to be made - I am comfortable with going ahead with
publication of the current paper (in spite of its preliminary nature) in view of the fact
publication of the paper may trigger follow-on research by other research groups as well.

Response: We appreciate the comments of the reviewer.

**Response to Referee #2 (Dr René Orth)**

R2C1: Review of Dongqin and Roderick "Inter-annual variability of the global terrestrial cycle" This study investigates the propagation of precipitation variability into the water cycle, i.e. into variations of runoff, evapotranspiration, and of storage changes. The authors show that this is mostly controlled by temperature (in wet regions), long-term aridity (in transitional regions), and by soil water storage capacity (in dry regions). Further, the results illustrate that the corresponding partitioning is different from the partitioning of mean precipitation into the means of these water cycle variables.

—————-

Recommendation: I think the paper requires major revisions.

The analysis is very interesting and provides new and fundamental insights into large- scale land surface hydrology. Related variability analyses are still not commonly done due to a lack of reliable data and underlying theory. This study can foster theory development in this area, and it underlines the importance of continuous improvement of the just-emerging global hydrological re-analysis datasets. Therefore I would be happy to see it published in HESS, but after some general revisions.

Response: We thank R2 for the evaluation and helpful comments on the manuscript.

R2C2: (1)   Next to the consideration of the soil water storage capacity and the mean temperature to explain variations in the partitioning of precipitation variability, I am missing the inclusion of vegetation type as an explanatory variable. It might have strong implications on evapotranspiration variability, and therefore also on runoff and storage variabilities.

Response: We agree with Dr René Orth that the inter-annual variability might be related to the other factors, e.g., vegetation type. However, given the fact that this is a new approach and the research is exploratory, we focused on relating the inter-annual variability with the most general hydrologic factors (i.e., the air temperature as a surrogate for snow/ice and water storage capacity). We expect to extend the current work to a more complete analysis (e.g., relation to vegetation) in future research and we hope others will follow by examining factors like vegetation since this will require the effort of many scientists.

R2C3: (2)   I agree with the authors that comprehensive hydrological reanalysis datasets are lacking, and the CDR dataset is an important contribution in that respect. Further, I appreciate the effort they make to validate the applicability of the dataset in the context of this study. However, also the CDR dataset is (necessarily) based on a model and hence it is not clear that the reported relationships are operating in nature, and not only in this model. To address this issue, I would like to see the key analyses from this study repeated with the state-of-the-art ERA5 reanalysis, which should be superior to ERA-Interim also in terms of land surface representation.

Response: As suggested by both R2 and the editor, we have compared the CDR ($P$, $E$, $Q$ and
$\Delta S$) with the same from the recently released ERA5. For this comparison, we use the same
1984-2010 period. We downloaded monthly $P$, $E$ and $Q$ (denoted as total runoff and calculated
by ERA5 as surface plus sub-surface runoff) from the ERA5 website. The water storage change
($\Delta S$) is not included in the ERA5 database, and we calculated it using mass balance for each
individual month during 1984-2010. We then conducted further analysis and found $P$ to be
similar in both CDR and ERA5 (Fig. R1). However, we found $E$ and (especially) $Q$ to be
generally **much** higher in ERA5 compared to CDR (Figs. R2-R3). This has important
consequences for the change in storage as described below.

[Figure]

Figure R1. Comparison of monthly precipitation $P$ between ERA5 and CDR databases. Top
panels (a) (b) show comparison of the mean monthly ($\bar{P}$) while bottom panels (c) (d) show
comparison of the standard deviation ($\sigma_P$) of monthly $P$.

[Figure]

Figure R2. The same as Fig. R1 but using monthly evapotranspiration *E* from ERA5 and CDR databases.

[Figure]

Figure R3. The same as Fig. R1 but using monthly runoff *Q* from ERA5 and CDR databases.

While the comparison with *P* (CDR vs ERA5 is reasonable, i.e., slope of the regression in Fig.
R1a = 1.0), we find that *E* from ERA5 is on average 25% larger (i.e. slope is 0.8, see Fig. R2a)
than *E* in CDR. Further, *Q* from ERA5 is on average 75% larger (i.e., slope is 0.57, see Fig.
R3a) than *Q* in CDR. Now we know that in CDR, the mass balance was enforced. The obvious
implication from these regressions is that in ERA5 the sum of *E* and *Q* must substantially
exceed *P*.

To further evaluate ERA5, we then integrated the monthly data to annual totals. Visually, the
results visually show similar global spatial patterns of long-term mean *P*, *E* and *Q* in the ERA5
database (see the Fig. R4a-c) to those in the CDR database (see Fig. 1 in the revised manuscript).
However, as noted above, the long-term mean annual water storage change ($\Delta S$, Fig. R4d)
implied by ERA5 showed evidence of a major problem with the local hydrology. In particular,
most regions of the earth surface show very large negative values for $\Delta S$, e.g., in the Amazon
long term mean annual $\Delta S$ is around -1000 mm. The implication is that over the 27-year period
(1984-2010), the annual storage change in ERA5 over the Amazon region is -1000 mm every
year and this equal 27 meters of storage change over the full period. This occurs in ERA5
because the sum of long-term mean annual *E* and *Q* is substantially greater than *P* in the
Amazon. This is physically not plausible. The same problem holds for many other warm
regions. In contrast, over the ice covered regions (e.g., Greenland), the hydrologic balance
implied a continuing gain in storage. Again, this is physically not plausible.

[Figure]

Figure R4. Mean annual (1984-2010) (a) *P*, (b) *E*, (c) *Q* and (d) $\Delta S$ in the ERA5 database.

Though the ERA5 is the state-of-the-art atmospheric reanalysis, we concluded that there was a
major problem with the hydrologic (mass) balance and that the "atmospheric-centric" ERA5
database was not yet suitable for use in hydrologic studies.

Returning to the suitability of the CDR database and its relation to the real world, there is ample
evidence that it is suitable for the analysis conducted here including:

| 195 | (i) | The enforcement of basic hydrologic concepts (mass balance). |
| 196 | (ii) | The numerous tests of CDR reported in the original Zhang et al 2018 HESS |
| 197 | | publication (that are summarized on lines 134-139 of the HESSD manuscript). |
| 198 | | Those tests include a (successful) comparison of CDR runoff to observations of |
| 199 | | monthly runoff at 165 medium size basins and 862 small basins. In fact, the |
| 200 | | assessment of CDR in the original paper was quite comprehensive as you would |
| 201 | | expect. |
| 202 | (iii) | We have augmented those extensive original tests by independently comparing |
| 203 | | monthly $E$ with FLUXNET tower data at 32 sites which confirmed that the CDR |
| 204 | | captured the general seasonal cycle in both $P$ and $E$ at those 32 sites (Fig. S3, S4, |
| 205 | | S5, Table S1 in the revised manuscript). We also used the same FLUXNET data to |
| 206 | | compare the variability in $P$ with variability in $E$ (Fig. S6 in the revised manuscript). |
| 207 | (iv) | We further compared CDR $E$ with two gridded $E$ databases that are not included in |
| 208 | | the source databases of CDR (LandFluxEval, MPI, see lines 159-166 in the revised |
| 209 | | manuscript and Fig. S7, S8) and the comparison was satisfactory. |
| 210 | (v) | We compared how the standard deviation for $E$ and the mean for $E$ are related in |
| 211 | | the CDR (Fig. 4 in the revised manuscript) and compared that with the same |
| 212 | | relations in LandFluxEval and MPI (Fig. S10 in the revised manuscript). Those two |
| 213 | | comparisons were satisfactory. |
| 214 | (vi) | The mean water cycle ($P$, $E$, $Q$) in CDR was shown to be consistent with the long- |
| 215 | | standing Budyko framework (Fig. 2 in the revised manuscript). |
| 216 | (vii) | The CDR data were consistent with the Koster & Suarez (1999) theory in the limit |
| 217 | | of sites that have limited water storage (Fig. 5 in the revised manuscript). |

That is a very comprehensive assessment.

Further, we readily acknowledge that the CDR database is the first hydrologic reanalysis and
we expect more 'hydrologic-centered' databases to compare it to in the near future. For that
reason we chose to only investigate the most general factors that we believe will stand the test
of time and we have also described the study as an initial exploratory survey at several places
in the manuscript.

R2C4: (3)   I appreciate the idea of investigating the influence of the soil water storage
capacity and the mean temperature on the variability partitioning. However, I think parts of the
conclusions drawn by the authors from Figures 8-10 are not supported by the data. For example,
I cannot see in Figure 10 that the temperature influence is particularly strong in very wet
regions. Rather, to me it seems to be strong in moderately wet and dry regions (Fig
10b,d,f,h,j,l,n,p). Further, also the aridity limit of 6 which the authors suggest in their
interpretation of the results in Figure 9, is arbitrary and not supported by the actual results.
Storage capacity is obviously having an influence already for aridity values above 2-3 (Fig.
9b,c,f,j,k). Overall, in these Figures there are many interesting patterns but the authors focus
only on few sub-plots and limit their interpretation to these. Therefore, I suggest to either show
less information/sub-plots there, or to develop explanations also for patterns emerging within
other sub-plots.

Response: We accept that Fig. 10 (Fig. 8 in the revised manuscript) is hard to interpret. On
reading the reviewers comments and going over the manuscript we realize the problem was
that we did not explicitly indicate the relevant panels (i.e., a, b, c, ….) and the text was not
well-formulated. This was an oversight correctly identified by the reviewer. In general, the data
in Fig. 10 was not particularly revealing (i.e., a negative result) but we actually focused the
discussion to the first and third columns but we did not identify them properly. In response, we
replaced the original text with the following (lines 307-314):

*"To understand the potential role of snow/ice in modifying the variance partitioning, we repeat*
*the previous analysis (Fig. 7) but here we use the mean annual air temperature ($\overline{T_a}$) to colour*
*the grid-cells to (crudely) indicate the presence of snow/ice (Fig. 8). The results are complex*
*and not easy to simply understand. The most important difference revealed by this analysis is*
*in the hydrologic partitioning between cold (first column) and hot (third column) conditions in*
*wet environments ($\overline{E_o}/\overline{P} \leq 0.5$). In particular, when $\overline{T_a}$ is high, $\sigma_P^2$ is almost completely*
*partitioned into $\sigma_Q^2$ in wet environments (e.g., $\overline{E_o}/\overline{P} \leq 0.5$, Fig. 8g). In contrast, when $\overline{T_a}$ is low*
*in a wet environment $\overline{E_o}/\overline{P} \leq 0.5$ in first column of Fig. 8), there are substantial variations in*
*the hydrologic partitioning. That result reinforces the complexity of variance partitioning in*
*the presence of snow/ice."*

R2C5: (4) The paper contains (too) many figures, which is diluting the main message(s), I
feel. For example, Figures 1 and 2 could be merged, Figure 5 could be moved to the
supplementary material, Figure 13 could be merged into Figure 8. The authors might have
further ideas to reduce the amount of figures. Moreover, I do not really understand the
difference between Figures 7 and 8, and why both are needed.

I do not wish to remain anonymous - René Orth.

Response: We respect the reviewer's opinion that we have too many figures – this is always a
hard balance to get right to everyone's satisfaction. We have moved the original Fig. 1 and Fig.
5 to the supporting material as suggested. There are now 12 figures in the revised manuscript
with another 12 in the supporting material. However, we do not think the original Fig. 13 (Fig.
11 in the revision) should be merged into original Fig. 8 (Fig. 6 in the revision) since the two
figures belong to different sections (original Fig. 8 for the relation between variance
partitioning and aridity section, original Fig. 13 for the case study section). Original Fig. 7
(Fig.5 in the revision) is a direct link to previous work while original Fig. 8 is the variance
partitioning in the CDR database. Hence while these two figures are similar, they make separate
independent contributions to the manuscript.

————————

Specific comments:

R2C6: line 8: Equation 2 not introduced yet line 13: It should be 'variabilities'.

Response: We have deleted the text 'Eq. 2' and changed 'the variability…' to 'that
variability…' to make the text clear to understand in the revised version of manuscript. Thanks.

R2C7: line 15: Some word is missing towards the end of the line

Response: We have checked line 15 in the original manuscript and did not find missing words?

R2C8: lines 35-39: Orth & Destouni (2018) might be relevant in this context and could be cited.

Response: The reference has now been cited in the revised manuscript.

R2C9: line 37: Not sure I get the point here.

Response: We mean that droughts and floods are typical extremes but that hydrologic variability encompasses more than just droughts and floods, i.e., hydrologic variability occurs across all time-space scales.

R2C10: lines 106-118: Please clarify that what you are determining here is actually not the soil water storage capacity, but rather the active range within which the soil moisture varies.

Response: Yes, exactly. We have modified the text and state the calculation to make this explicit in Lines 108-110 in the revised manuscript: "*For the storage, the active range of the monthly water storage variation was used to approximate the water storage capacity ($S_{max}$).*".

R2C11: lines 157-163: I would recommend to replace the LandFluxEVAL and the Jung et al. datasets with more recent gridded ET datasets such as the Jung et al. 2019 dataset and the GLEAM dataset (Martens et al. 2017).

Response: The reason we chose the LandFluxEVAL and MPI databases is that they are among the most widely used and validated $E$ data that were also **not used** to develop the CDR database. We do not think adding a comparison to the latest GLEAM database would be as useful since an earlier version of GLEAM (v2a) was actually an input to the data assimilation scheme used to construct the CDR (see Table 1 in Zhang et al., 2018, HESS). Instead, the more appropriate approach would be to revise the CDR data assimilation but incorporating the latest GLEAM database but that is well beyond the scope of this work. (Also see R2C3 for similar comments about ERA.) We could replace the MPI we used with the updated database (Jung et al., 2019) but we do not see how that would alter the results.

R2C12: line 180: Gudmundsson et al. (2016) might be relevant in this context and could be cited.

Response: The reference has been cited in the revised manuscript. Thanks.

R2C13: line 181: What is meant by seasonality here? I thought you are considering annual data? In general, I think the considered temporal and spatial scales and resolution need to be more clearly stated and motivated at the beginning of the manuscript. Also, the role of these decisions on the results could be discussed.

Response: Yes, we are using annual data. But we know that differences of the intra-year seasonal timing (phase) of precipitation and $E_o$ do have an effect on the annual water balance (as per the seminal work by Chris Milly in the early 1990s.). To make this more clear, we have added a statement in in the revised manuscript (Lines 100-101): "*In this study we focus on the inter-annual variability and the monthly water cycle variables (P, E, Q and ΔS) to annual totals.*"

Given the initial stage for this type of research and our plan to include the seasonal variations in future work (also see R1C2 and R1C3), a statement has been added in the revised manuscript (Lines 505-508): "*That result demonstrates that deeper understanding of the process-level interactions that are embedded within each of the three covariance terms (e.g., the role of seasonal vegetation variation) will be needed to develop process-based understanding of variability in the water cycle in these biologically productive regions (0.5<$\overline{E_o}/\overline{P}$ <1.5).*".

R2C14: line 252/253: I could not find this discussion in section 5!? Would be important to explain these discrepancies, though.

Response: Thanks for pointing this oversight out. The underlying scientific issue here is that the original Koster and Suarez (1999) work assumed negligible water storage change. In that sense the original results of Koster and Suarez (1999) can be seen as an upper limit and any variance in storage can only reduce the partitioning of variability in $P$ to variability in $E$ under dry conditions (Fig. 7). We have added a short discussion on this in the revised manuscript (Lines 488-492): "*This result explains the overestimation of $\sigma_E/\sigma_P$ by the empirical theory of Koster and Suarez (1999) which implicitly assumed no inter-annual change in storage. The Koster and Suarez empirical theory is perhaps better described as an upper limit that is based on minimal storage capacity, and that any increase in storage capacity would promote the partitioning of $\sigma_P^2$ to $\sigma_{\Delta S}^2$ particularly under dry conditions (Figs. 10-12).*".

R2C15: line 327 & 333: 'leaving very limited variance' - not really true given your statement in lines 385-387

Response:

We show the $P$, $E$, $Q$ and $\Delta S$ time series along with the relevant variances and covariances in Fig. 12. Starting with the two dry sites, at the site with low storage capacity (Site 1), the time series shows that $E$ closely follows $P$ leaving annual $Q$ and $\Delta S$ close to zero (Fig. 12a). The variance of $P$ ($\sigma_P^2 = 206.9$ mm$^2$) is small and almost completely partitioned into the variance of $E$ ($\sigma_E^2 = 196.9$ mm$^2$), leaving very limited variance for $Q$, $\Delta S$ and all three covariance components (Fig. 12b). At the site with high storage capacity (Site 2), $E$, $Q$ and $\Delta S$ do not simply follow $P$ (Fig. 12c). As a consequence, the variance of $P$ ($\sigma_P^2 = 2798.0$ mm$^2$) is shared between $E$ ($\sigma_E^2 = 1150.2$ mm$^2$), $\Delta S$ ($\sigma_{\Delta S}^2 = 800.5$ mm$^2$) and their covariance component ($2cov(E, \Delta S) = 538.4$ mm$^2$, Fig. 12d). Switching now to the remaining wet and hot site (Site 3), $Q$ closely follows $P$, with $\Delta S$ close to zero and $E$ showing little inter-annual variation (Fig. 12e). The variance of $P$ ($\sigma_P^2 = 57374.4$ mm$^2$) is relatively large and almost completely partitioned into the variance of $Q$ ($\sigma_Q^2 = 57296.4$ mm$^2$), leaving very limited variance for $E$ and $\Delta S$ and the three covariance components (Fig. 12f). We also examined numerous other sites with similar extreme conditions as the three case study sites and found the same basic patterns as reported above.

The text here refers to the site-based case studies (line 327 – Fig. 12a (Fig. 10 in the revised manuscript) – Site 1; line 333 – Fig. 12 f – Site 3) while the later text (lines 385-387) refers to the general pattern across all grid-boxes, i.e., Fig. 4 (Fig. 3 in the revised manuscript). We have corrected this misunderstanding by rewriting lines 385-387 (lines 470-478 in revised manuscript) to indicate the relevant figures as follows:

*"With that in mind, we were surprised that the inter-annual variability of water storage change*
*($\sigma^2_{\Delta S}$) is typically larger than the inter-annual variability of evapotranspiration ($\sigma^2_E$) (cf. Fig.*
*3b and 3d). The consequence is that $\sigma^2_{\Delta S}$ is more important than $\sigma^2_E$ for understanding inter-*
*annual variability of global water cycle. A second important generalisation is that unlike the*
*variance components which are all positive, the three covariance components in the theory*
*(Eq. 2) can be both positive and negative. We report results here showing both large positive*
*and negative values for the three covariance terms (Fig. 3efg). This was especially prevalent*
*in biologically productive regions ($0.5 < \overline{E_o}/\overline{P} < 1.5$, Fig. 3eg)."*

R2C16: lines 403-405: I cannot see this from Figure 8.

Response: Agreed. That was our mistake. The reference to Fig. 8 (Fig. 6 in the revised
manuscript) should be to Fig. 4 (Fig. 3 in the revised manuscript, global pattern of water cycle
variability) and we have revised that in the revision.

R2C17: Section 5: Overall a bit lengthy with too much summarizing, I think. Could be shorter,
and more concise.

Response: Both R2 & R3 (see R3C4) had divergent views about the summary sections of our
original manuscript.

After looking at both comments (R2, R3) and the structure of the original manuscript, we
concluded that the original Discussion and Conclusions sections were repetitive and not well
formulated.

In response, we have combined the original sections into a single section 5 (Discussion and
Conclusions) and have streamlined the text accordingly. We believe that this has substantially
improved the manuscript.

R2C18: Figure 3: Why are there data points outside the physically plausible range?

Response: We assume you mean points with *E* exceeding *P*? This is possible in for example,
regions with run-on, or irrigation. We have further investigated those points and also find that
some of them come from the parts of Greenland that had not been masked out (Fig. 1).

R2C19: Figure 4: Many values seem to be cut at 10 as this is the end of the color bar. You
could use log scale here for the color bar.

Response: Yes, the scale for *P* in Fig. 4a (new Fig. 3a in the revised manuscript) is saturated
with the maximum value of the color bar 10,000. The reason we chose 10,000 as the limit was
to show the patterns for both the relatively high (e.g., $\sigma^2_P$, $\sigma^2_Q$ and $\sigma^2_{\Delta S}$) and low variabilities
(e.g., $\sigma^2_E$, $2\text{cov}(E, \Delta S)$) while keeping the same scale for all panels. We have tried to modify
this figure by using a log scale (see Fig. R5) to mitigate saturation, but it made the spatial
patterns very difficult to distinguish compared with Fig. 3 in the revised manuscript (original
Fig. 4) especially for the covariance panels (Fig. R5e-g). Therefore, we thought it better to keep
the original legend in Fig. 3.

[Figure]

Figure R5. Water cycle variances ($\sigma_P^2$, $\sigma_E^2$, $\sigma_Q^2$, $\sigma_{\Delta S}^2$) and covariances ($cov(E,Q)$, $cov(E,\Delta S)$, $cov(Q,\Delta S)$). Note that we have multiplied the covariances by two (see Eq. 2).

[Figure]

Figure 3 (original Figure 4). Water cycle variances ($\sigma_P^2$, $\sigma_E^2$, $\sigma_Q^2$, $\sigma_{\Delta S}^2$) and covariances ($cov(E,Q)$, $cov(E,\Delta S)$, $cov(Q,\Delta S)$). Note that we have multiplied the covariances by two (see Eq. 2).

R2C20: References:

Gudmundsson, L., P. Greve, and S. I. Seneviratne, 2016: The sensitivity of water avail- ability to changes in the aridity index and other factorsâ˘T˘A probabilistic analysis in the Budyko space, Geophys. Res. Lett. 43 (13), 6985-6994.

Jung, M., S. Koirala, U. Weber, K. Ichii, F. Gans, G. Camps-Valls, D. Papale, C. Schwalm,
G. Tramontana, and M. Reichstein, 2019: The FLUXCOM ensemble of global land-
atmosphere energy fluxes. Scientific Data, 6 (74).

Martens, B., D. G. Miralles, H. Lievens, R. van der Schalie, R. A. M. de Jeu, D. Fernández-
Prieto, H. E. Beck, W. A. Dorigo, and N. E. C. Verhoest, 2017: GLEAM v3: satellite-
based land evaporation and root-zone soil moisture, Geosci. Model Dev. 10, 1903–1925.

Orth, R., and G. Destouni, 2018: Drought reduces blue-water fluxes more strongly than green-
water fluxes in Europe. Nature Communications, 9, 3602, doi: 10.1038/s41467- 018-06013-7

Response: We appreciate Dr René Orth for listing all the reference mentioned above in the
comments, and we have read and cite these reference accordingly in the revised manuscript.
Thanks.

**Response to Referee #3 (Anonymous)**

R3C1: This study tries to partition the inter-annual variability in precipitation (P), i.e., the source term in terrestrial water cycle, into variabilities in three sink terms in terrestrial water cycle (ET, Q, ΔS), and then to relate the partitioning of variabilities to various factors like temperature, aridity, and storage capacity. I think this type of study at global scale is rather new, if not first of its kind at global scale, and thus very interesting to the hydrology community. This is the case mostly because there has been a lack of "hydrologic reanalysis" (CDR) for such kind of analysis in the first place. At the same time, this effort couldn't fully answer many of the questions set forth at the beginning, leaving perhaps "more questions than answers" (as phrased by another referee). The authors have done a solid amount of thorough analysis and experiments toward the questions of interest and these analyses are also well designed too.

Overall I consider this manuscript of good quality, both scientifically and technically, and thus publishable in HESS with several concerns addressed.

Response: We agree that this is a first-of-its-kind study and thank the referee for the encouraging positive comments on the manuscript.

R3C2: My primary concern is there is a lack of general "signal-to-noise" discussions to better inform readers to what extent the findings are significant signals from the underlying data (CDR, Zhang et al., 2018) and how much of it could be due to data uncertainties (or possible artifacts due to how the data is produced). For example, the ET products that went into the CDR (satellite products, reanalysis, etc.) share some similarity in their production methods (e.g., Penman-Monteith or Priestley-Taylor type of schemes). Such similarity may limit the variability of ET in CDR. Of course, the plants do apply a strong filter on the inter-annual variability based on their survival need. Such uncertainty analysis may be difficult but I think some qualitative and general assessment would be very beneficial.

Response: The CDR uses a formal data assimilation scheme based on mass balance that weights the various inputs, and thereby produces uncertainty estimates for each variable ($P$, $E$, $Q$, $\Delta S$). The original paper (Zhang et al., 2018 HESS) includes a formal assessment of the sensitivity of $P$, $E$, $Q$ over large regions (continents, basins) using the coefficient of variation (see original Figures 2, 3, 4, 5, 6, 7 in Zheng et al., 2018 HESS). We actually followed from that work and used those uncertainty estimates (lines 124-132 in the revised manuscript) to identify and mask out regions where the uncertainty was large relative to the magnitude of the fluxes. This screening procedure removed most grid-boxes from the Himalayas, Sahara Desert and Greenland (see Fig. S2 in the revised manuscript).

Secondly, while it is true that some of the products might share similarity in producing, for example, $E$ (Penman-Monteith, Priestley-Taylor as the examples noted by the reviewer) the data assimilation is a comprehensive approach that includes all available estimates of $P$, $E$, $Q$ and $\Delta S$ at each grid box. With mass balance enforced, the CDR estimates represent a composite product that is designed to avoid bias of the type described by the reviewer as much as possible by using all available estimates of the hydrologic fluxes. As we have described in a response to Reviewer 2 (see R2C3), the CDR has been extensively validated in the original publication.

In that context, our goal was not to assess the CDR, but rather to use it for this "first-of-a-kind" study on the sources and sinks of inter-annual hydrologic variability. We have added words at the end of the manuscript that we expect further improvement and validation of obtained patterns (Lines 459-460): "*Further, we expect future improvements and modifications as the hydrologic community seeks to further develop and refine these new reanalysis databases.*".

R3C3: Also, at the scale of the CDR (0.5 degree), I would say the partitioning is more complicated than just a result of several factors. The horizontal transport of water, seasonality, local water use, etc., can add a lot of noise. I wouldn't say it is not possible to do it at 0.5 degree, but it would probably be less noisy at a slightly coarser scale. Also, there could be much more controlling factors for the partitioning than being investigated, e.g., land cover/land use, LAI, topography.

Response: We agree with the reviewer that the partitioning is complex and could be related to the other factors, e.g., land cover/land use, LAI and horizontal transport of water due to topography, etc. In this first-of-a-kind analysis we chose to focus on the zero'th order physical factors (storage capacity, snow/ice) at the CDR data resolution (0.5 degree), but we fully expect more detailed analysis to follow, e.g., vegetation plant-based variables as discussed by the reviewer. We have added new text in the last paragraph of section 4.5 that speculates on the important role of vegetation processes that addresses this comment by R3. We have also emphasized that again in the final concluding paragraph of the manuscript.

R3C4: Finally, given that this study does tend to raise more questions than answers, I feel the authors should provide some more insights on what we can do from the analysis and findings in this study. What can we do with the numbers concluded here? Validating models? Improving single models like Budyko? Hydrologic/water risk analysis? Climate system behavior/sensitivity and hydrologic impacts of climate changes? And how can we improve our understanding in the future? What kind of new data at what scales would be critical to answering such questions? I feel this paper is incomplete without offering some of such insights.

Response: Please also see R2C17.

In further response, we have modified the final paragraph to set out a rough guideline for future research (lines 511-515): "*
[revised manuscript text omitted]

| | Deleted: 4 |
| Deleted: Fig. 8 |

**Response to Editor**

Dear authors, thank you very much for the revised version of your manuscript. Since all other reviewers suggested minor revisions, I only requested one of the reviewers (René Orth) to comment on the new version.

He appreciates the additional analyses performed, but still feels that some of the variables have not been validated, such as run-off. The reviewer suggests additional analyses, regarding (1) use updated version of the Jung et al. dataset (Jung et al., 2019) and (2) use the E-RUN dataset (Gudmundsson and Seneviratne, 2016) to validate the runoff.

I find that the manuscript has much improved and you have made good effort addressing the reviewers comments. I think the manuscript is almost ready for publication, given some amendments. In view of the fact that the Jung et al. (2019) paper was published only after the submission of the manuscript (although the data were available earlier), I will not insist on this additional analysis. However, please consider the Gudmundson and Seneviratne (2016) dataset. Please also attend to the detailed comments of the reviewer.

I am looking forward to your resubmitted manuscript,

Anke Hildebrandt.

Response: We thank the editor for the evaluation and comment on the revised manuscript. As suggested by the editor and reviewer, we have conducted additional analyses using the E-RUN database (Gudmundson and Seneviratne, 2016) and the FLUXCOM database (updated version of MPI database, Jung et al., 2019). We also revised the manuscript accordingly as well as conducted a point-by-point response to all the comments by the reviewer.

The main comment here is a further cross-validation of the CDR runoff based on the E-RUN database. The comparison results show that both the long-term mean ($\bar{Q}$) and standard deviation ($\sigma_Q$) of the monthly runoff in the E-RUN database are very similar with those in the CDR database. We further added the comparison results of runoff in the revised Supplementary Material, and also changed the text accordingly in the revised manuscript. Please also see R2C3 for a detailed response to this point.

Another comment is about using the FLUXCOM database instead of MPI in the validation of the CDR evapotranspiration $E$. As has been noted by the editor, the FLUXCOM database paper was published after the submission of this manuscript. In addition, the monthly FLUXCOM data is currently only available (open to public) for a much shorter period (2001-2010) compared with both the monthly CDR (1984-1010) and the original MPI (1982-1011) databases. As strongly suggested by R2, we conducted further comparison between the CDR and FLUXCOM databases, and the results are similar with those comparison between the CDR and MPI databases. Given the limited time period in the FLUXCOM database and the similarity of comparison results using the MPI and FLUXCOM databases, we choose to keep the results of the MPI database in the Supplementary Material. Please also see R2C2 for detailed response.

Again, we sincerely appreciate both the editor and reviewer for constructive suggestions and comments on the revised manuscript.

**Response to Referee #2 (Dr René Orth)**

R2C1: Second review of Yin and Roderick "Inter-annual variability of the global terrestrial cycle"

The paper has overall improved as the authors have addressed many of the concerns raised by me and the other reviewers. However, one important issue, and several minor points remain unresolved.
* * *
Response: We thank Dr René Orth for the evaluation and helpful comments on the revised manuscript. Please see detailed response to all the comments as follows.

R2C2: Main comment: As mentioned in my previous review, I think it is critical for this study to show that the discovered patterns are not just implemented in the model used to derive the CDR dataset. It has to be shown that similar patterns are present across independent datasets, as only this can indicate that nature is indeed operating this way. I appreciate efforts in this direction made by the authors, namely the consideration of the LandFlux-EVAL dataset, the Jung et al. dataset, and the ERA5 reanalysis. But I believe that these analyses need to be expanded before the paper can be published:

(1) I understand that the authors do not want to use GLEAM as a reference dataset as this was used in the derivation of the CDR reanalysis. But instead the Jung et al. dataset should be updated to the 2019 version (Jung et al. 2019). The authors stated in their response: 'We could replace the MPI we used with the updated database (Jung et al., 2019) but we do not see how that would alter the results.' This is not about altering the results, but about using state-of-the-art alternative datasets to illustrate the robustness of the CDR-based results. I do not see the point in using an almost 10-year old dataset while updated and much evolved datasets exist.

Response: As suggested by R2, we conducted further comparison between the CDR and FLUXCOM ($0.5° \times 0.5°$, monthly, 2001-2010) (updated version of MPI database, Jung et al., 2019) databases, and the results are shown in Fig. R1. The results are similar with the previous comparison between the CDR and MPI databases, showing underestimation of the monthly mean $E$ and bias and scaling offset in the standard deviations of monthly $E$ in the CDR database compared with the FLUXCOM database.

However, currently the monthly FLUXCOM database is only available (open to public) for the restricted period 2001-2010, which is much shorter than both the CDR (available during 1984-2010) and the MPI (available during 1982-2011) databases. Given the limited time period for the FLUXCOM database and the similar comparison results of the MPI and FLUXCOM to the CDR databases, we propose to keep the results based on the original MPI database in the Supplementary material.

[Figure]

**Figure R1. Comparison of monthly evapotranspiration $E$ between FLUXCOM and Climate Data Record (CDR) databases. Top panels (a) (b) show comparison of the mean monthly ($\bar{E}$) while bottom panels (c) (d) show comparison of the standard deviation ($\sigma_E$) of monthly $E$.**

R2C3: (2) I also appreciate the ERA5-based analyses which the authors have done in response to my previous comments. I share their conclusion that this dataset is not suitable to be used in the context of this study. However, this way the runoff results remain not confirmed with independent data. Therefore I suggest to use the E-RUN gridded runoff dataset (Gudmundsson and Seneviratne 2016) for this purpose.

I do not wish to remain anonymous - Rene Orth.
* * *
Response: As suggested, we conduct further comparison of the monthly runoff between the E-RUN ($0.5° \times 0.5°$, monthly, 1951-2015) (Gudmundsson and Seneviratne, 2016) and CDR databases. The comparison is conducted based on the overlap of time (1984-2010) and space (Europe) in both databases, and the results are shown in Figs. R2-R3. We can see that both the long-term mean ($\bar{Q}$) and standard deviation ($\sigma_Q$) of the monthly runoff show very similar spatial patterns in the E-RUN and CDR databases (Fig. R2). The grid-by-grid comparison also shows close agreement (Fig. R3). We have added these results to the revised Supplementary Material (Figs. S10-S11), and also added the text in the revised manuscript as follows (lines 165-169): "*The comparison of runoff Q between the E-RUN and CDR databases show that the two databases have very similar spatial patterns of both the long-term mean ($\bar{Q}$) and standard deviation ($\sigma_Q$) of the monthly Q (Fig. S10). The grid-by-grid comparison results are also encouraging, showing slight bias of both the long-term mean and standard deviation of monthly Q in the CDR database compared with the E-RUN database (Fig. S11).*".

[Figure]

Figure R2. Mean ($\overline{Q}$) and standard deviation ($\sigma_Q$) of monthly runoff $Q$ in the E-RUN and Climate Data Record (CDR) databases in the area of spatial overlap (Europe). Top panels (a) (b) show the mean monthly ($\overline{Q}$) while bottom panels (c) (d) show the standard deviation ($\sigma_Q$) of monthly $Q$.

[Figure]

Figure R3. Comparison of monthly runoff $Q$ between the E-RUN and Climate Data Record (CDR) databases in the area of spatial overlap (Europe). Top panels (a) (b) show comparison of the mean monthly ($\overline{Q}$) while bottom panels (c) (d) show comparison of the standard deviation ($\sigma_Q$) of monthly $Q$.

R2C4: lines 53-55: This statement somewhat ignores the efforts leading to the ERA-Land (Balsamo et al. 2013) and MERRA-Land (Reichle et al. 2011) datasets.

Response: We have acknowledged the efforts in developing land-based products in the revised manuscript by modifying the sentence to (lines 53-57): "*Though efforts have been taken to develop land-based products from atmospheric reanalyses, e.g., ERA-Land (Balsamo et al. 2013) and MERRA-Land (Reichle et al. 2011) databases, however, the central aim of atmospheric re-analysis is to estimate atmospheric variables. That atmospheric-centric aim, understandably, ignores many of the nuances of soil water infiltration, vegetation water uptake, runoff generation and many other processes of central importance in hydrology.*". The relevant reference has also been cited in the revised manuscript.

R2C5: line 75: 'the various ... databases' - after only reading the text up to this point it is not clear what is meant here

Response: It means the databases used in this study will be introduced and described in Section 2. To make this sentence more clear, we have modified it in the revised manuscript as follows (lines 77-79): "*We begin in Section 2 by describing the various climate and hydrologic databases used in this study, and also include a further assessment of the suitability of the CDR database for this initial variability study.*".

R2C6: line 78: it should be 'these variabilities'

Response: Done. Thank you.

R2C7: lines 88/89: 'in time step t' - these are all fluxes which are accumulated during time steps t-1 and t; also, I would mention here that the time step considered in this study is 1 year

Response: We have added the annual time step in this sentence to make it more clear in the revised version (lines 90-91): "*with P the precipitation, E the evapotranspiration, Q the runoff and ΔS the total water storage change in time step t (annual in this study).*". Thank you.

R2C8: lines 91/92: 'Eq (1) is the familiar...' - this sentence is an unnecessary repetition

Response: This sentence has been deleted in the revised manuscript. Thanks.

R2C9: line 96: known here?

Response: To make the meaning of this sentence more clear, we have removed the word 'known' and modified it in the revised manuscript as follows (lines 98-99): "*We use the Climate Data Record (CDR) database (Zhang et al., 2018) which is a recently released global land hydrologic re-analysis.*".

R2C10: line 103: The SRB dataset only extents until 2007 (if I am not mistaken) while the analyses in this study consider a time period until 2010. How can you still use the SRB data then?

Response: The aim of this study is to investigate the inter-annual variability of global water cycle based on the CDR database, which extends from 1984 to 2010. During the construction process, the CDR database made some assumptions considering the 27-year period (1984-2010) as an integrity, e.g., the long-term (27-year) storage change to be zero. To better investigate the inter-annual variability by using the CDR database in this study, we choose to stick to the CDR period, i.e., 1984-2010.

While the SRB database is only available from 1984 to 2007 (not to 2010), we only use it to calculate the long-term $E_\mathrm{o}$ ($\overline{E_o}$) and further estimate the aridity index ($\overline{E_o}/\overline{P}$). We believe the three-year period difference would not have a material impact on the aridity index estimation or change the general conclusions in this study. Thanks.

R2C11: lines 105/106: Sentence is hard to understand, please rephrase.

Response: We have rephrased this sentence in the revised manuscript as follows (lines 107-108): "*In general, we anticipate two important factors, i.e., the water storage capacity and the presence of ice/snow at the surface, which are most likely to have influence on the partitioning of hydrologic variability.*". Thanks.

R2C12: line 160: Please comment on the offset.

Response: We have added more details for the offset and modified the sentence in the revised manuscript as follows: "*In terms of variability, the standard deviations of monthly E from the CDR are in very close agreement with the LandFluxEVAL database (Fig. S7c), but there is a bias and scaling offset for the comparison with the MPI database particularly for the grid-cells with low standard deviation of E (Fig. S8c).*".

R2C13: line 177: I would replace 'trend' with 'pattern'

Response: Done. Thank you.

R2C14: line 180: not clear what is meant here with 'physics of runoff generation'

Response: Yes, we agree that the 'physics of runoff generation' is not clear and we have replaced it with more specific term in the revised manuscript as follows (lines 187-189): "*However, there is substantial scatter due to, for example, regional variations related to seasonality, water storage change and the landscape characteristics*".

R2C15: lines 178-181: Padron et al. 2017 is relevant in this context, and could be cited.

Response: The reference has now been cited in the revised manuscript. Thank you.

R2C16: lines 188, 203, 223: 'very different' is not obvious to me from the comparison of Figs 1 and 3. Please clarify.

Response: Here we mean it is very different between the partitioning of $\bar{P}$ and $\sigma_P^2$. In brief, the $\bar{P}$ is mostly partitioned into $\bar{E}$ or $\bar{Q}$. However, for the partitioning of $\sigma_P^2$, $\sigma_E^2$ is generally very small with $\sigma_P^2$ mostly partitioned into $\sigma_Q^2$, $\sigma_{\Delta S}^2$ and even the covariance components. Please see the more comprehensive and detailed analyses in the revised manuscript (lines 199-209).

R2C17: lines 225-226: This is an important finding which should be highlighted in the abstract and/or conclusions.

Response: Yes, the finding here has been added in the abstract (lines 11-12): "==Instead we find that $\sigma_P^2$ is mostly partitioned between $\sigma_Q^2$, $\sigma_{\Delta S}^2$ and the associated covariances with limited partitioning to $\sigma_E^2$.==".

R2C18: lines 294-303: If the main conclusion is that things are complex, and there is no particular lesson learned here, then I would suggest to remove this section. It confuses readers and distracts from the relevant main messages of the study.

Response: While the results here are complex and not easy to understand, we still could have some implications obtained here, for example, the difference between partitioning of $\sigma_P^2$ at high and low temperature. That difference does show the important role of temperature in the partitioning of $\sigma_P^2$, which might be helpful for the future studies. Therefore, we would like to keep this section in the revised manuscript.

R2C19: lines 307-328: It feels inconsistent that in addition to the wet and hot grid cell no wet and cold grid cell has been selected as a case study (as was done in the case of high and low water storage capacity).

Response: The reason we did not pick any case study site here is because there is substantial scatter in wet and cold conditions ($\overline{E_o}/\bar{P} \leq 0.5$ in first column of Fig. 8). The partitioning of $\sigma_P^2$ in wet and cold conditions is so complex that no grid-cell could be chosen as a representative case study site. Instead of a case study site, we further illustrate the importance of snow/ice presence in variance partitioning (lines 425-426) and expect more emphasis on this in the future studies that our manuscript will inspire.

R2C20: - While this study is performed at annual time scales, the authors could add some outlook/clarification that the revealed variability propagation across the water cycle might behave differently at shorter time scales

Response: Yes, we agree that the variability partitioning might be different at various time scales. In response, we have added an expectation for future work at various time scales in the revised manuscript (lines 408-410): "*These general principles of variance partitioning in the water cycle above may vary at different time scales (e.g., monthly, daily), and we expect more details of the variability partitioning across various temporal scales to be investigated in future studies.*".

R2C21: - Figures 2,5, and others display physically implausible values - please comment on this

Response: Yes, there are some grid-cells showing physically implausible values in Figs. 2 and 5. In this study, we have tried to exclude the grid-cells with high uncertainty (please see Section 2.3 and Fig. S2), therefore, it is unlikely that those implausible values are caused by data uncertainty/error. While checking the location of those grid-cells, we found that they almost appear in/close to the Greenland. Therefore, we guess those physically implausible values are caused by the permanent ice/glacier. As also noted in this study, with the presence of snow/ice, it is very complex in the variance partitioning. In this study, we highlighted regions with snow/ice coverage. We except future studies to further uncover the role of snow/ice in the variance partitioning and show details of these physically implausible values.

R2C22: - It is not intuitive that non-consistent (logarithmic/non-lagarithmic) axes are used for E0/P across different figures.

Response: Yes, the axes for the aridity index ($E_o/P$) are linear in Figs. 2, 5 and 6 and logarithmic in Figs. 7 and 8. The underlying reason for that is because there are different purposes in presenting the results in these figures. In Figs. 2, 5 and 6, we show the relation of long-term mean and variance to $E_o/P$. It is better to use the regular non-logarithmic axes to compare with results in previous studies (e.g., Budyko-curve and Koster and Suarez analyses) that also use linear axes. While in Figs. 7 and 8, we highlight the role of storage capacity and physical phase (solid/liquid) in variance partitioning in both extremely dry and wet environments. We found the logarithmic axes to better show the necessary details in Figs. 7 and 8.

R2C23: References:

Balsamo, G., C. Albergel, A. Beljaars, S. Boussetta, E. Brun, H. Cloke, D. Dee, E. Dutra, J. Muñoz-Sabater, F. Pappenberger, P. de Rosnay, T. Stockdale, and F. Vitart, 2013: ERA Interim Land: a global land water resources dataset. Hydrol. Earth Syst. Sci., 19, 389–407.

Gudmundsson, L., and S.I. Seneviratne, 2016: Observation-based gridded runoff estimates for Europe (E-RUN version 1.1). Earth Syst. Sci. Data, 8 (2), 279–295.

Jung, M., S. Koirala, U. Weber, K. Ichii, F. Gans, G. Camps-Valls, D. Papale, C. Schwalm, G. Tramontana, and M. Reichstein, 2019: The FLUXCOM ensemble of global land-atmosphere energy fluxes. Scientific Data, 6 (74).

Padron, R.S., L. Gudmundsson, P. Greve, and S.I. Seneviratne, 2017: Large-Scale Controls of the Surface Water Balance Over Land: Insights From a Systematic Review and Meta-Analysis. Water Res. Resour., 53 (11), 9659-9678.

Reichle, R.H., R.D. Koster, G.J.M.D. Lannoy, B.A. Forman, Q. Liu, S.P.P. Mahanama, and A. Toure, 2011: Assessment and enhancement of MERRA land surface hydrology estimates. J. Clim., 24, 6322–6338,

Response: We thank Dr René Orth for listing all the reference in the comments, and we have read and cite these reference accordingly in the revised manuscript.

[revised manuscript text omitted]